# Quantifying negative selection in human 3′ UTRs uncovers constrained targets of RNA-binding proteins

Scott D. Findlay ◉[1], Lindsay Romo[1,2] & Christopher B. Burge ◉[1] ✉

Many non-coding variants associated with phenotypes occur in 3′ untranslated regions (3′ UTRs), and may affect interactions with RNA-binding proteins (RBPs) to regulate gene expression post-transcriptionally. However, identifying functional 3′ UTR variants has proven difficult. We use allele frequencies from the Genome Aggregation Database (gnomAD) to identify classes of 3′ UTR variants under strong negative selection in humans. We develop intergenic mutability-adjusted proportion singleton (iMAPS), a generalized measure related to MAPS, to quantify negative selection in non-coding regions. This approach, in conjunction with in vitro and in vivo binding data, identifies precise RBP binding sites, miRNA target sites, and polyadenylation signals (PASs) under strong selection. For each class of sites, we identify thousands of gnomAD variants under selection comparable to missense coding variants, and find that sites in core 3′ UTR regions upstream of the most-used PAS are under strongest selection. Together, this work improves our understanding of selection on human genes and validates approaches for interpreting genetic variants in human 3′ UTRs.

Since the sequencing of the human genome, identifying functional genetic variants that influence human phenotypes, including disease has been a central goal. For variants lying in protein-coding exons, the genetic code aids greatly in interpretation. However, the vast majority of candidate causal variants emerging from genome-wide association studies (GWAS) lie outside of protein-coding regions[1–3], and likely impact a variety of regulatory elements, making interpretation much more challenging[4,5].

Since transcription is a major point of regulation for gene expression, much of the search for functional non-coding variants has focused on transcriptional regulation and regions upstream of the coding sequence[6–8]. Other regions such as 3′ untranslated regions (3′ UTRs) have been less explored, despite playing major roles in post-transcriptional regulation involving cleavage and polyadenylation, mRNA stability, mRNA localization and translation[9]. Previous studies have found that 3′ UTRs explained GWAS genotyped SNP heritability at a rate 5-fold higher than expected[10] and that expression quantitative trait loci (eQTLs) were more enriched (> 2-fold) in 3′ UTRs than any other non-coding annotation analyzed[11], and hundreds of pathogenic / likely pathogenic 3′ UTR variants have been submitted to ClinVar, demonstrating an abundance of impactful genetic variation in 3′ UTRs.

Mechanistically, interactions between RNA-binding proteins (RBPs) and their target RNAs lie at the heart of virtually all post-transcriptional gene regulation in 3′ UTRs. As examples, Argonaute proteins guided by cellular microRNAs (miRNAs) bind most mRNAs to repress expression[12], cleavage and polyadenylation specificity factors (CPSFs) bind polyadenylation signals (PASs) to define 3′ ends of transcripts[13–15], and Pumilio family proteins (PUM1/PUM2) and AU-rich element (ARE) binding proteins such as TIA1 regulate stability and translation of bound transcripts[16–19]. Thus, the collection of RBP binding sites in 3′ UTRs constitutes a set of non-coding elements enriched for regulatory activity.

Recent large-scale efforts using techniques such as enhanced crosslinking and immunoprecipitation (eCLIP) have characterized RBP

[1]Department of Biology, Massachusetts Institute of Technology, Cambridge, MA 02142, USA. [2]Present address: Boston Children's Hospital, Boston, MA 02115, USA. ✉e-mail: cburge@mit.edu

binding sites throughout the transcriptome[20], providing a basis for the discovery of allele-specific RNA-RBP interactions[21,22]. However, these studies are inherently limited to the small number of variants that are heterozygous in the cell lines used, and little work has been done to more broadly assess the evolutionary pressures acting on variants that modulate RBP-RNA interactions. The premise of such an approach is that the number of times variant alleles are observed in human populations will be lower within constrained functional regulatory elements in the genome, leaving a signature of negative/purifying selection during the "natural experiment" of human evolution. Such efforts were initially limited in resolution by the number of genomes with genome-wide and deep sequencing data available[23,24]. More recently, projects such as the genome aggregation database (gnomAD) and UK Biobank have cataloged genetic variation from tens of thousands of whole genomes[25,26]. The gnomAD Consortium also developed the Mutability-Adjusted Proportion Singleton (MAPS) metric that summarizes the allele frequency spectrum across collections of variants to quantify negative selection. This metric improved on previous measures by capturing non-selective but well-known forces impacting allele frequency spectra, such as differential mutability, that can greatly confound the evaluation of negative selection[25,27–30].

While this type of approach has been applied to identify genetic variation under strong selection in non-coding regions such as 5′ UTRs[31] and introns at splice sites[32,33], it has not yet been broadly applied to 3′ UTRs, suggesting that signals of negative selection may be challenging to uncover in these regions. Instead, the few instances where negative selection in 3′ UTRs has been inferred have been limited in scope, did not adjust for mutability, or were secondary to other efforts[34–37].

In this work, we detail patterns of negative selection across diverse classes of regulatory elements in human 3′ UTRs. We introduce the intergenic MAPS (iMAPS) approach that is well-suited to detect signals of negative selection in non-coding regions of the transcriptome. Using this method, we confirm a major role for RBP-RNA interactions in shaping the 3′ UTR regulatory landscape by describing numerous classes of genetic variants that are under strong selection, influence transcript levels, and can improve interpretation of non-coding genetic variants.

## Results

To quantify the extent to which different classes of 3′ UTR variants are under negative selection, we used their allele frequency spectra (AFS). For interpretability, these AFS must be calibrated by comparing them to the AFS from a more neutrally evolving region of the genome to establish a baseline level of negative selection. The extent to which there is a shift toward rare variants in the AFS of the variants of interest indicates the level of negative selection. Previous work quantifying negative selection in broadly defined genomic regions has typically relied on calibrating the allele frequencies of variants of interest to synonymous coding variants. Some previous approaches have matched variants based on base change and flanking dinucleotide context, as an additional critical aspect of calibration is to control for the influence of mutability on the allele frequency spectrum[25], although other work has not explicitly calibrated at this resolution[34]. We reasoned that these approaches may not be sufficient for assessing more specific classes of non-coding regulatory variation where sequence composition is often less complex[38] and the magnitude of negative selection is likely more modest. Furthermore, some synonymous variation is clearly under selection in connection to RNA splicing, mRNA stability, etc., so calibrating to a more neutrally evolving class of variation is desirable. We developed a rigorous method to better account for the influence of mutability on the allele frequency spectrum by 1) calibrating to variation in intergenic regions, 2) considering additional sequence context, and 3) considering the impact of transcription on DNA repair and mutation (Fig. 1a) for the reasons given below.

First, the vast majority of RBPs that bind predominantly in 3′ UTRs also bind coding sequence extensively[20], so calibrating to a transcribed region may dampen or obscure signals of negative selection from such regulatory elements. Relative to synonymous variants in transcribed coding regions, intergenic variants are under less selection[25] and are devoid of RNA regulatory elements characteristic of 3′ UTRs such as RBP binding sites.

Second, there are many more intergenic variants available than synonymous coding variants (over 30-fold more, even after conservative filtering), providing greater statistical power to account for the influence of sequence (and other) contexts beyond dinucleotide composition that have been shown to substantially affect mutability[29]. Using intergenic variants, we identified many such contexts that extended up to five bases on each side of the variant. The amount of extended sequence context considered was not constant, as data was limiting for many extended contexts (see Methods). This approach captured significant influences of extended nucleotide contexts across all base changes, affecting the majority (54%) of all 3′ UTR variants, relative to a dinucleotide-only approach (Methods, Fig. 1b).

Third, previous work has performed calibration using paired dinucleotide contexts, grouping together reverse complement contexts such as C[A > G]T and A[T > C]G[25,27,31]. However, it is known that strand-biased processes including transcription-associated mutagenesis and transcription-coupled repair influence mutability in transcribed regions for some sequence contexts, e.g., A > G/T > C mutation rates are higher when A > G occurs on the coding strand[39–41]. This effect has not commonly been incorporated into mutation rate models[29] and has not yet been accounted for in any analyses of negative selection. We assessed the influence of transcription on mutability in 3′ UTRs using de novo mutation data[42] in addition to allele frequency data from gnomAD, where increased mutability presents as a decrease in the proportion of variants that are singletons. Consistent with previous work[39], such effects were substantial for A > G/T > C base changes and minimal for other contexts (Fig. 1c). Variable (and often unknown) rates of germline transcription across genes make it difficult to account for the influence of transcription in any calibration approach. Therefore, we excluded A > G/T > C variants from our analysis, where transcriptional effects are most apparent.

To address these issues, we developed the intergenic MAPS or "iMAPS" approach, an extension of the MAPS metric[25,27,31], to quantify negative selection in noncoding regions including 3′ UTRs. Overall, iMAPS more extensively accounts for non-selective factors influencing allele frequency spectra (as summarized by proportion singleton values), which resulted in improved calibration compared to MAPS (Supplementary Fig. 1). We benchmarked 3′ UTR iMAPS values against canonical classes of coding variation. As a whole, negative selection in 3′ UTRs was higher than in intergenic regions and slightly lower than for synonymous coding variants, paralleling the pattern of evolutionary conservation across mammals in these regions[43]. More specifically, we looked at regulatory element annotations in 3′ UTRs, including eCLIP peaks marking RBP binding sites[20], miRNA target sites predicted by TargetScan+ (all targets of miRNA families conserved in mammals or more broadly)[44], and the canonical polyadenylation signal hexamer AWUAAA (W = A or U). In aggregate, none of these individual annotations had iMAPS scores exceeding that of synonymous variants (Fig. 1d), suggesting that this conventional approach of one-dimensional annotation intersection was insufficient to enrich for variants under negative selection and that variant interpretation in 3′ UTRs may require more nuanced approaches. We reasoned that there are subsets of variants within these annotations under strong selection and hypothesized that further stratification of variants aided by additional annotations could uncover these classes of deleterious variants in 3′ UTRs.

We first focused on general RBP binding sites in 3′ UTRs, with the goal of identifying precise/short RBP binding sites of high confidence

using complementary orthogonal methods: for each RBP with available data, we identified the highest affinity RBPamp motif (based on high throughput in vitro RNA Bind-n-Seq (RBNS) data[38,45] in the vicinity of each of the more than 25,000 ENCODE eCLIP peaks[20] and termed these sites "RBPamp eCLIP-Proximal" or "ReP" sites (Fig. 2a). At either 10 or 11 bases long, ReP sites were about five-times more precise than typical eCLIP peaks. The significant enrichment of ReP sites around the 5′ ends of eCLIP peaks highlights the coherence of these datasets and validates this approach to identify precise RBP binding sites marked by eCLIP peaks for the vast majority of available RBPs (Supplementary Fig. 2). We found that variants in ReP sites were up to 6-fold enriched for variants that altered transcript levels in a 3′ UTR massively parallel reporter assay (MPRA[46]; Fig. 2b). Additionally, relative to those in position-matched eCLIP peak regions, positions within ReP sites were up to 65% more likely to be conserved across species (Supplementary Fig. 3). Together, these data suggest many ReP sites are RNA elements with endogenous RBP-binding and regulatory capacities.

Next, we leveraged RBPamp affinity models to predict the impact of each gnomAD variant within ReP sites on RBP binding. Variants where the derived allele was predicted to have substantially reduced affinity to the RBP relative to the ancestral allele were termed "disrupting" (or "lost"), while variants where the derived and ancestral

alleles were predicted to have similar affinity were termed "preserving" (Methods). Here, we emphasized ReP sites that we term "focal," where most of the local affinity to the RBP is concentrated in the ReP site itself. We reasoned that variants in focal sites would be more likely to have large and predictable impacts on regulation relative to variants in sites where the affinity is spread across tens of nucleotides, since the binding location can be more confidently inferred, and there is less potential for redundant regulation when affinity is concentrated to one short element. Indeed, for 3′ UTR ReP site variants included in an MPRA measuring transcript levels[46], variants in focal ReP sites conferred significantly larger alterations in transcript levels relative to variants in non-focal ReP sites (Supplementary Fig. 4).

Both disrupting and preserving variants were identified in ReP sites spanning over a dozen diverse RBPs with established regulatory roles in 3′ UTRs (Fig. 2c). We observed that disrupting variants were subject to significantly stronger negative selection than preserving variants, supporting the importance of this distinction. Furthermore, disrupting variants in ReP sites of increasingly high affinity were under increasingly strong negative selection. Above the highest minimum affinity analyzed (30% of an ideal binding site), the negative selection experienced by these variants exceeded the genome-wide average of missense coding variants (Fig. 2c). Conversely, for all minimum

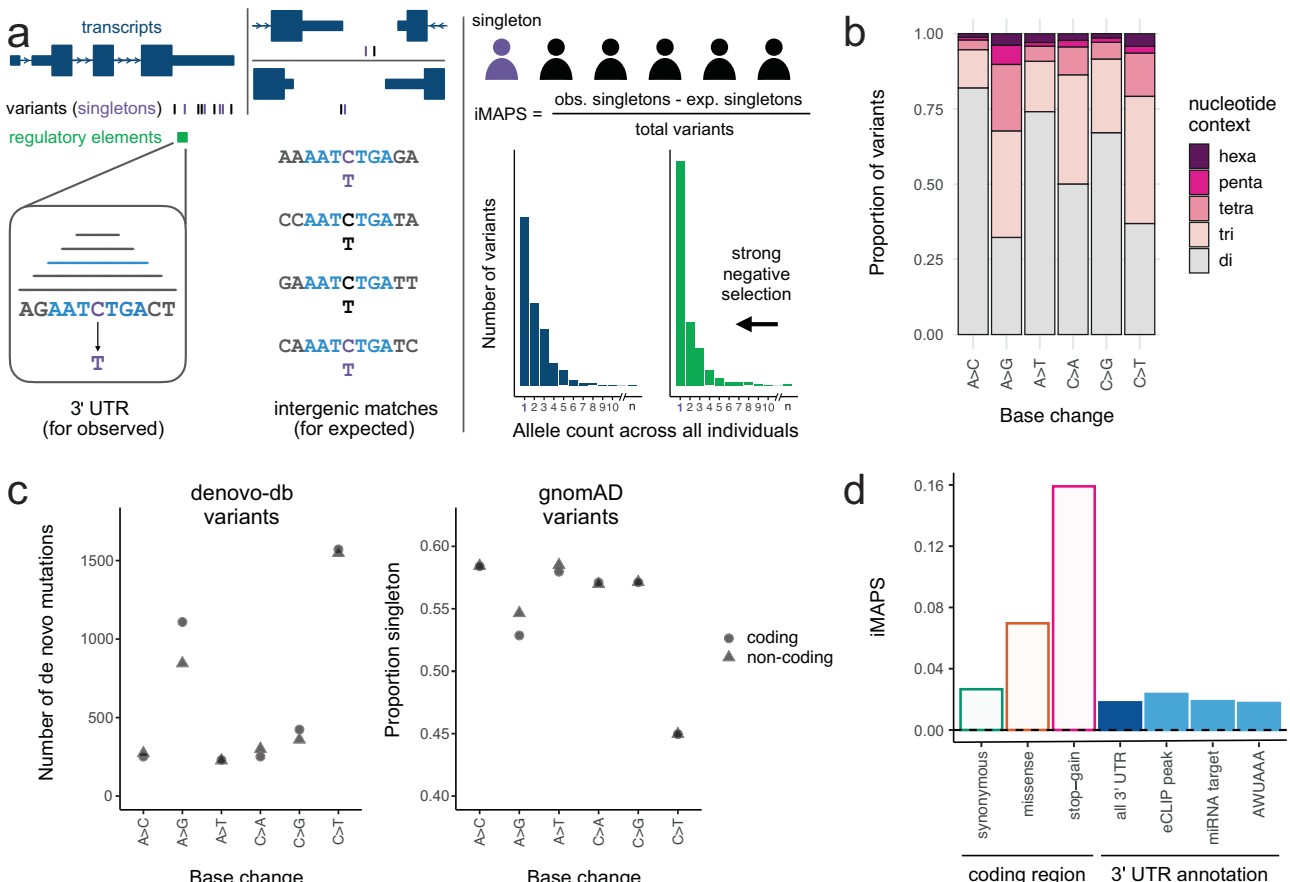

**Fig. 1 | Using iMAPS to quantify negative selection in 3′ UTRs. a** Schematic of iMAPS approach. Left: each 3′ UTR variant is matched to multiple intergenic variants with the same base change and flanking sequence context (ranging from dinucleotide to hexanucleotide). Right: Top: the number of singletons (variant alleles detected once across all individuals in gnomAD; depicted in purple) is used to calculate iMAPS. "obs" = observed, exp. = expected. Bottom: sets of variants under strong selection (green) have an excess of singletons and thus higher iMAPS scores. **b** The proportion of 3′ UTR variants for which considering extended nucleotide contexts when matching to intergenic variants significantly affects the expected singleton rate. Total number of 3′ UTR variants for each level of sequence context: hexa = 324,986; penta =

309,244; tetra = 1,347,534; tri = 3,351,380; di = 4,591,107. **c** Left: A > G/T > C mutations have the highest strand bias in de novo mutation rate relative to all other base changes. Right: accordingly, the proportion of variants that are singleton is also the most strand biased for A > G/T > C variants, in the expected direction. "Coding" (circles) and "non-coding" (triangles) indicate the strand of the labeled base change. In B and C, each base change pair (different base change on each strand) is labeled with the pair member where the ancestral base is A or C. Note: some pairs of points are completely overlapping at the resolution shown. **d** Summary of overall average negative selection for 3′ UTR variants relative to intergenic variants (dashed line at 0), benchmarked against CDS variants that differentially affect encoded protein.

affinities analyzed, preserving variants did not surpass the much lower average negative selection experienced by synonymous coding variants (Fig. 2c). Using orthogonal eQTL data, we also found that ReP site variants with increasingly higher PIP values (representing the

probability of causing an expression difference) were more likely to be disrupting than preserving (Fig. 2d). We also tested ReP site-disrupting 3′UTR variants in a parallel reporter assay in cells, where 39% (11/28) of the variants tested reproducibly modulated steady-state transcript

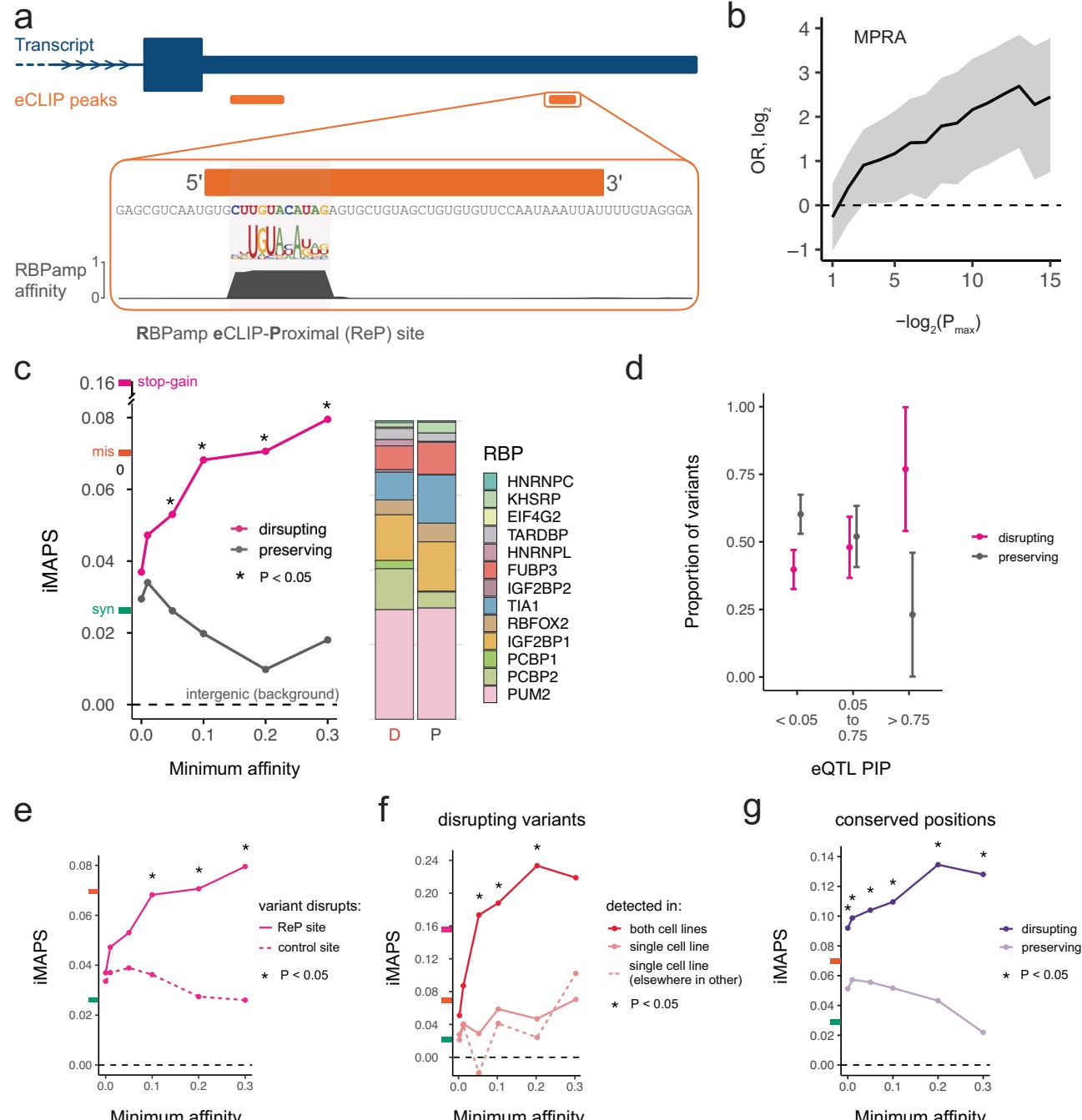

**Fig. 2 | Variants disrupting ReP sites in 3′ UTRs are under strong negative selection. a** A ReP site is the highest affinity RBP binding site at an eCLIP peak for that RBP. **b** ReP site variants are enriched for significant transcript abundance modulating activity in a massively parallel reporter assay (MPRA) of 3′ UTR variants[46]. Data are presented as odds ratios with 95% confidence intervals. *P* values were obtained from Griesemer et al.[46]. OR = odds ratio. $P_{max}$ = *P* value significance threshold. **c** Variants disrupting RBP affinity / binding at ReP sites are under stronger selection than those that preserve affinity / binding. The stacked bar plot shows the proportion of variants found within each RBP's ReP sites. "D" = disrupting, "P" = preserving. iMAPS scores for synonymous (green), missense (orange), and/or stop-gain (pink) coding variants are shown on y-axis here and in e-g for reference. **d** High PIP (likely causal) eQTL variants in ReP sites are much more

likely to be disrupting than preserving. Data are presented as proportions with error bars indicating 95% confidence intervals. **e** Variants disrupting ReP sites are under stronger selection than those disrupting the control site with closest affinity in each gene. The dashed line at 0 indicates background levels of intergenic selection. **f** Variants disrupting ReP sites detected by eCLIP in both cell lines are under stronger selection than variants disrupting ReP sites detected in only one cell line. Dashed line is data for a control set of ReP sites detected in both cell lines but at different positions in the same 3′ UTR. **g** At highly conserved positions (phastCons 100-way = 1.0), variants disrupting ReP sites are under stronger selection than those that preserve RBP affinity/binding. Asterisks (*) indicate one-sided Fisher Exact Tests with *P* < 0.05. Exact *P* values and the number of variants available for analysis are included in Supplementary Data 1.

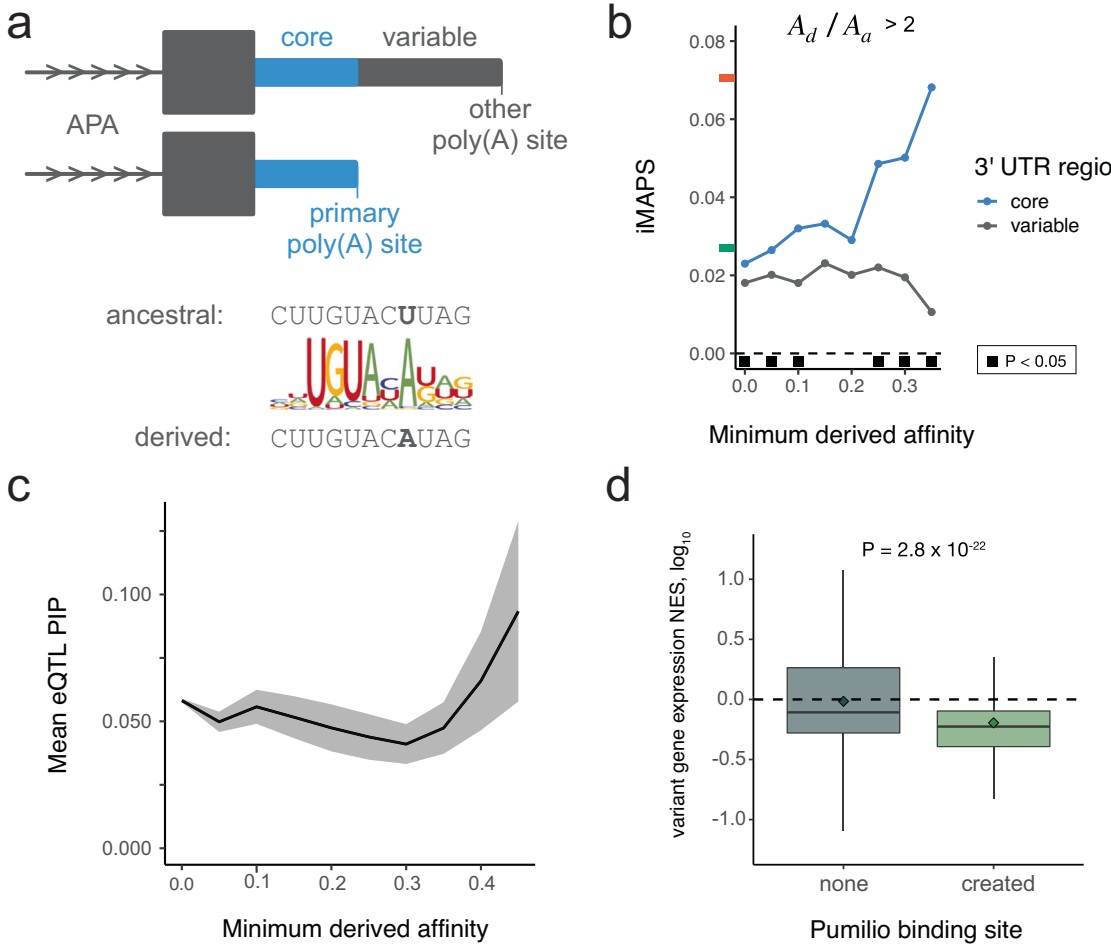

**Fig. 3 | The impact of de novo Pumilio binding sites created by genetic variants.** **a** Top: Illustration of the "core" 3′UTR region (blue) upstream of the primary (most utilized) poly(A) site and the "variable" 3′UTR region (gray) present only in isoforms resulting from use of secondary poly(A) sites. Bottom: Schematic illustrating an example of Pumilio binding site creation. **b** Variants creating Pumilio binding sites in core 3′UTR regions are under strong negative selection. $A_d$ = derived affinity, $A_a$ = ancestral affinity. Black squares indicate one-sided Fisher Exact Tests with $P < 0.05$. Exact $P$ values and the number of variants analyzed are included in Supplementary

Data 1. **c** eQTL variants that create high-affinity Pumilio sites typically have higher PIP values. Shaded area indicates mean +/- standard error. **d** eQTL variants creating Pumilio sites ($n = 448$ eQTLs) are also associated with decreased transcript levels relative to those with no site ($n = 70,318$ eQTLs), consistent with the destabilizing impact of Pumilio binding. The $P$ value shown is the result of a one-sided Wilcoxon Rank Sum test. NES = $\log_{10}$ normalized effect size. Diamonds indicate mean NES values. Lines within boxes indicate median NES values. Boxes extend from first quartile to third quartile values. Whiskers extend to 1.5x the interquartile range.

levels (Supplementary Fig. 5). Collectively, these data demonstrate that RBPamp can be used to prioritize RBP binding site variants more likely to drive deleterious gene expression changes, accelerating variant interpretation efforts and functional variant identification.

We considered whether biases related to gene expression might contribute to the magnitude of the above negative selection results. Since statistical power for eCLIP peak-calling in each gene is a function of eCLIP read counts for that gene, eCLIP peaks and therefore ReP sites are enriched in relatively highly expressed genes (Supplementary Fig. 6), which tend to be more constrained. For example, we found coding variants from genes with 3′UTR ReP sites to have slightly elevated synonymous and missense iMAPS (~0.034 and 0.10, respectively) relative to genome-wide averages. However, when considering a set of control sites matched for gene expression and RBP affinity with ReP sites, we found that variants that disrupt ReP sites had substantially higher iMAPS scores than variants that disrupted these control sites (Fig. 2e). This analysis shows that the presence of eCLIP signal at ReP sites is an important feature in predicting constraint, and that independent of eCLIP, high RBP affinity and high gene expression are not generally sufficient to identify sites under strong selection.

In an effort to better understand other aspects of RBP binding sites in relation to negative selection, we focused on RBP binding events shared across cell lines, and on cross-species sequence conservation. Since eCLIP was performed in two distinct cell lines for many RBPs, we reasoned that ReP sites identified in both cell lines would be enriched for RBP binding sites that are broadly utilized across cell types and might therefore be involved in regulation of fundamental cellular processes. Indeed, we found that variants disrupting ReP sites present in both cell lines were under very strong negative selection (Fig. 2f). This was not simply a result of 3′UTR eCLIP peaks detected in both cell lines deriving from more highly expressed genes (where increased read depth provides increased statistical power to call peaks), as disruption of a control set of ReP sites from genes where ReP sites were detected in both cell lines but at distinct 3′UTR positions was much less deleterious. These results suggest that detection of eCLIP signal at the same site across distinct cell types enriches for functional binding.

When restricting to positions conserved across vertebrates, variants disrupting ReP sites were highly deleterious. Importantly, variants at the same sites that preserve RBP binding were under much weaker selection, supporting that preservation of RBP binding is a

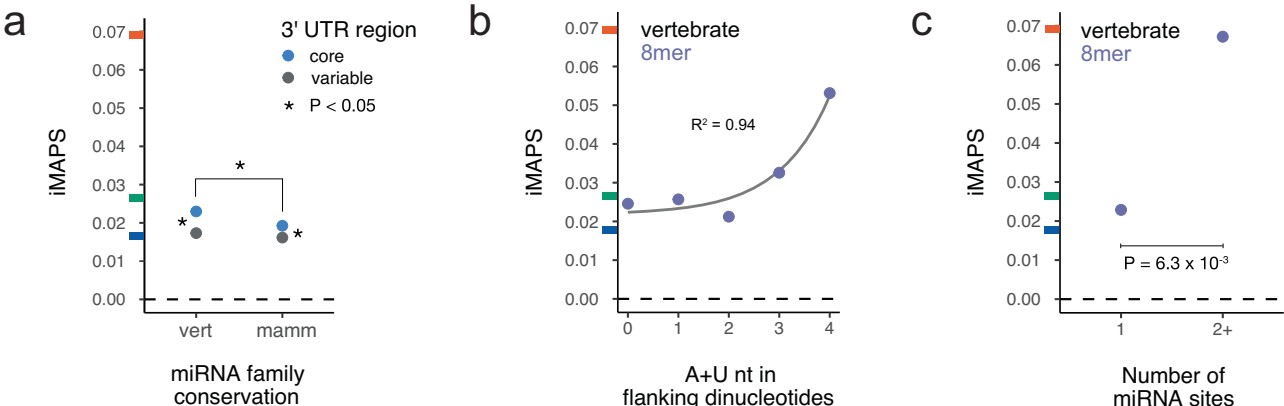

**Fig. 4 | Negative selection at miRNA target sites in 3′ UTRs. a** miRNA targets (including 8mer, 7mer-m8, and 7mer-A1 target types) are under selection in core 3′ UTR regions for both broadly conserved ("vert" = vertebrate) and conserved ("mamm" = mammalian) microRNA families. $n = 179{,}220$ variants for vertebrate core, 197,573 for mammalian core, 137,103 for vertebrate variable, and 147,679 for mammalian variable. Asterisks (*) indicate one-sided Fisher Exact Tests with $P < 0.05$. Vertebrate $P = 4.3 \times 10^{-15}$. Mammalian $P = 1.6 \times 10^{-5}$. **b** Stronger selection is observed for 8mer targets with increasing AU content of the dinucleotides immediately preceding and following the target site. The gray line is a linear model describing the data using an exponential fit. $R^2$ shows Pearson's product-moment correlation. **c** Variants within targets recognized by two or more broadly conserved miRNA families are under stronger selection than those within targets of a single miRNA family. $n = 10{,}912$ variants for single targets and $n = 389$ variants for variants in targets of multiple miRNAs. A one-sided Fisher Exact Test $P$ value is shown. Tick marks on the y-axis indicate the genome-wide average iMAPS for 3′ UTR (blue), synonymous (green), and missense (orange) coding variants. The dashed line at 0 indicates background levels of intergenic selection.

major contributor to the high conservation observed at these positions (Fig. 2g). This finding also highlights the capacity of our approach — unlike most measures of cross-species sequence conservation — to capture divergent selection signals based on the nature of the base change introduced by the variant. Overall, these results demonstrate that ReP sites, combining in-cell eCLIP and in vitro RBNS/RBPamp binding site data, reveal precise and functional regulatory sites bound by RBPs.

Moving beyond genetic variants within ancestral binding sites, we next explored whether derived variant alleles that create new RBP binding sites experience substantial selective pressure. While eCLIP signals likely arise specifically from derived alleles in some cases, detecting a reasonable number of such events would require conducting eCLIP across far more individuals and cell lines than are available. In the absence of such data, we expected that it might be difficult to uncover strong signal for the creation of RBP binding sites. Therefore, we focused on creation or strengthening of binding sites to Pumilio family proteins (PUM1 and PUM2) since these RBPs have well characterized binding sites, bind with high affinity (sub-nM $K_d$), and can strongly reduce target gene expression via transcript destabilization[18].

Since this and all subsequent analyses involve regulatory elements predicted based on primary sequence alone, it was important to consider the alternative polyadenylation (APA) context. Specifically, we distinguished between variants in "core" and "variable" 3′ UTR regions for each gene. Core regions are upstream of the most utilized or "primary" poly(A) site according to the average reads per million across all tissues and cell lines in polyA_DB[47]. Variable regions are downstream of the primary poly(A) and are included in processed transcripts only when a less frequently selected poly(A) site is utilized (Fig. 3a).

Applying this approach to the creation of RBP binding sites, we found that variants increasing Pumilio affinity are increasingly deleterious for derived alleles with higher affinities, and creation of sites at least 35% as strong as an ideal Pumilio site approached the strength of selection seen for missense coding variants. This strong signal was specific to the creation of putative binding sites within core 3′ UTR regions. Destabilization in core regions is expected to impact the bulk of transcripts from a gene. Conversely, variants creating putative

binding sites within variable 3′ UTR regions did not exceed levels of selection seen for synonymous coding variants (Fig. 3b). Overall, 3′ UTR gnomAD variants generate putative Pumilio sites ($A_d / A_a > 2$ and $A_d > 0.25$) across 2,658 protein-coding genes. Using independent eQTL data, we verified that variants creating high-affinity Pumilio binding sites had higher probabilities of being causal of gene expression differences (Fig. 3c) and were more likely to decrease gene expression (Fig. 3d). Thus, in addition to disruption of RBP binding being deleterious, creation of strong RBP binding sites is under negative selection, at least in some contexts.

Since regulation by miRNAs is a widespread form of post-transcriptional gene regulation, we investigated negative selection at miRNA target sites (8mer, 7mer-m8, and 7mer-A1) in 3′ UTRs according to TargetScan[44]. Targets of both "broadly conserved" (vertebrate-wide) and "conserved" (mammalian-wide) miRNA families were under stronger negative selection in core 3′ UTR regions upstream of primary polyadenylation sites compared to variable regions, which are present in a subset of transcripts. Targets of more broadly conserved miRNAs were generally under stronger selection (Fig. 4a), consistent with the finding that miRNAs conserved in mammals and not more broadly have fewer conserved targets[48]. As a positive control, conserved target sites of these miRNA families were under stronger selection than non-conserved sites, as has been previously observed[49], exceeding average levels for synonymous coding variants (Supplementary Fig. 7).

We next sought to explore whether certain subsets of miRNA target sites were under stronger selection, absent target conservation across species. We focused on the most potent target sites: 8-mer targets of broadly conserved miRNA families. Within these sites, we observed a positive association between AU-rich dinucleotides flanking target sites and negative selection (Fig. 4b). The regions flanking miRNA targets are enriched for adenosine[50], and targets in AU-rich contexts are known to confer stronger binding and increased repression, likely due to increased target site accessibility[51–53]. We also found that overlapping targets of more than one miRNA family were under stronger negative selection (Fig. 4c), presumably because such overlap increases the likelihood that the site has regulatory activity.

To assess negative selection associated with polyadenylation, we focused on the polyadenylation signal (PAS), which is the major determinant of poly(A) site selection[54–57]. We focused specifically on

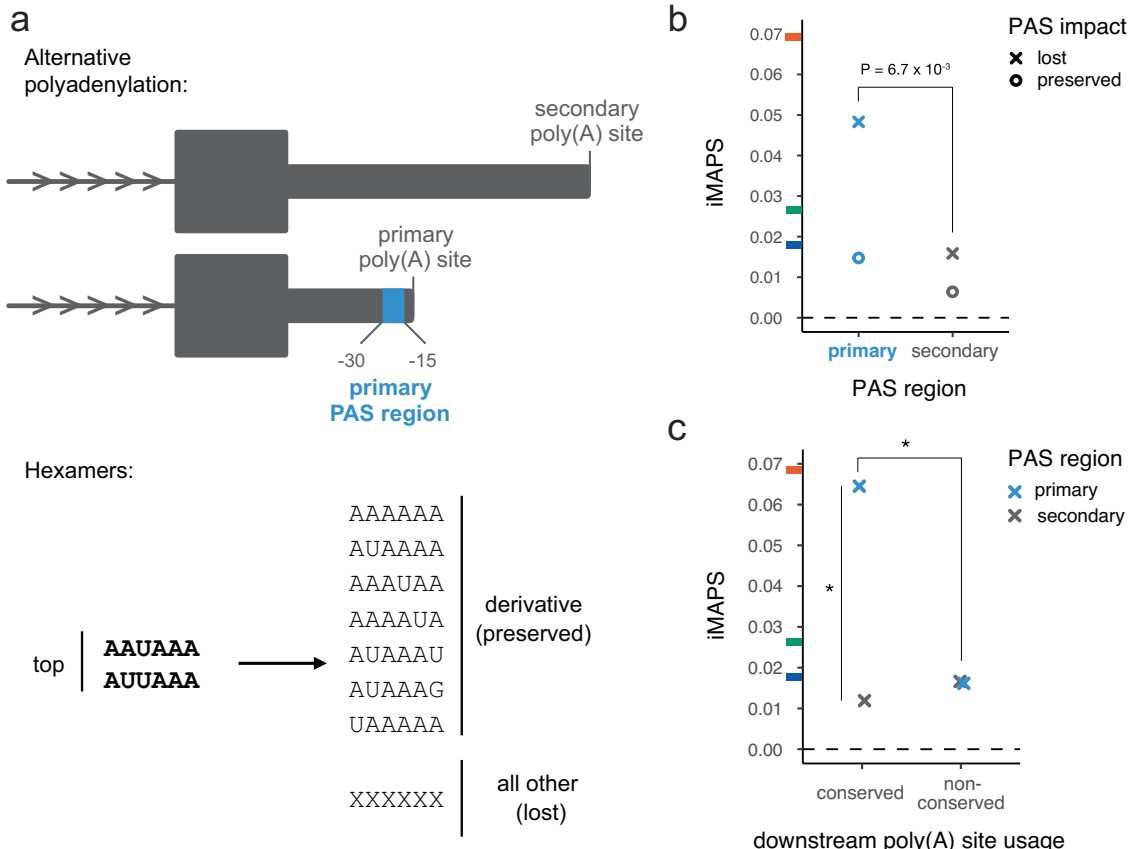

**Fig. 5 | Negative selection on PASs in 3′ UTRs. a** Schematic of polyadenylation sites in 3′ UTRs. Top: the primary PAS region for a given gene (associated with the most commonly used poly(A) site) is shown in blue. AWUAAA hexamers found in any other regions (gray) are considered secondary. Bottom: Canonical / top PAS hexamers match AWUAAA. A PAS is considered preserved if the variant results in a canonical or derivative PAS hexamer, otherwise it is considered lost. **b** PAS loss in primary PAS regions is under stronger selection than PAS lost in secondary regions. $n = 1,571$ variants for lost PAS in primary regions and $n = 14,867$ variants for lost PAS in secondary regions. A one-sided Fisher Exact Test $P$ value is shown. **c** PAS loss in primary regions is under stronger selection when the downstream poly(A) site is also utilized in mouse or rat (i.e. "conserved"). $n = 1041$ variants for primary conserved, 2928 for secondary conserved, and 530 for primary non-conserved. Asterisks (*) indicate one-sided Fisher Exact Tests with $P < 0.05$. Primary & conserved vs. secondary & conserved $P = 0.0016$. Primary & conserved vs. primary & non-conserved $P = 0.035$. Tick marks on the y-axis indicate the genome-wide average iMAPS for 3′ UTR (blue), synonymous coding (green), and missense coding (orange) variants. The dashed line at 0 indicates background levels of intergenic selection.

variants in the two "top" PAS hexamers matching AWUAAA. We used data from the polyA_DB database[47] to classify AWUAAA hexamers as "primary" (30 to 15 bases upstream of the most utilized poly(A) site in a gene) or "secondary" (at any other 3′ UTR position). Analogous to our analysis of RBP motifs, PASs were classified as "lost" if the overlapping variant altered the AWUAAA motif without creating any other enriched hexamer from Ni et al.[58], and were otherwise classified as "preserved" (Fig. 5a).

We found that variants causing primary PAS loss were under stronger selection than variants causing secondary PAS loss. Conversely, preserving variants experienced selection similar to background 3′ UTR levels (Fig. 5b). We next assessed selection in the context of "conserved" human poly(A) sites, defined as those that have a homologous active poly(A) site in mouse and/or rat[47]. We found that, in aggregate, virtually all of the selection against variants disrupting primary PASs was attributable to PASs associated with these conserved poly(A) sites. Only background 3′ UTR levels of selection were observed for disruption of primary PASs associated with non-conserved poly(A) sites (Fig. 5c). Our results are consistent with a recent analysis arguing that most secondary poly(A) sites are non-adaptive[59], allowing for some exceptions, of course[60].

Having identified classes of genetic variation under strong selection acting in different modes of post-transcriptional gene regulation in 3′ UTRs, we sought to summarize the frequency of such variants.

This analysis was done in a manner that controlled for biases in gene expression between datasets such that different iMAPS results could be compared. We labeled variants belonging to classes with iMAPS ≥ 0.06 (approaching the genome-wide average for missense coding variants) "highly disruptive," and those belonging to classes with iMAPS ≥ 0.03 (greater than the genome-wide average for synonymous coding variants) "moderately disruptive." We use the term "disruptive" based on the strong inference that these variants disrupt molecular interactions with regulatory function, analogous to how missense variants disrupt protein coding. These labels will be helpful in efforts to interpret genetic variation associated with disease, but alone are insufficient as evidence of pathogenicity[61]. We were able to label over 5,000 gnomAD variants in 3′ UTRs (> 1 out of every 2,000) as highly disruptive and almost 20,000 gnomAD variants in 3′ UTRs (> 1 out of every 600) as moderately disruptive (Fig. 6a). Furthermore, we have identified all possible single nucleotide variants in 3′ UTRs that would be considered highly disruptive based on our classifications and have included the complete catalog of 140,000+ such variants (as well as additional variants) in Supplementary Data 2.

While most of our classes did not exceed missense level selection, this is not to suggest that there is an upper limit on selection in 3′ UTRs. For example, we have demonstrated that variants at conserved positions disrupting ReP sites are under very strong selection (Fig. 2g) but chose not to include classes based on cross-species conservation in

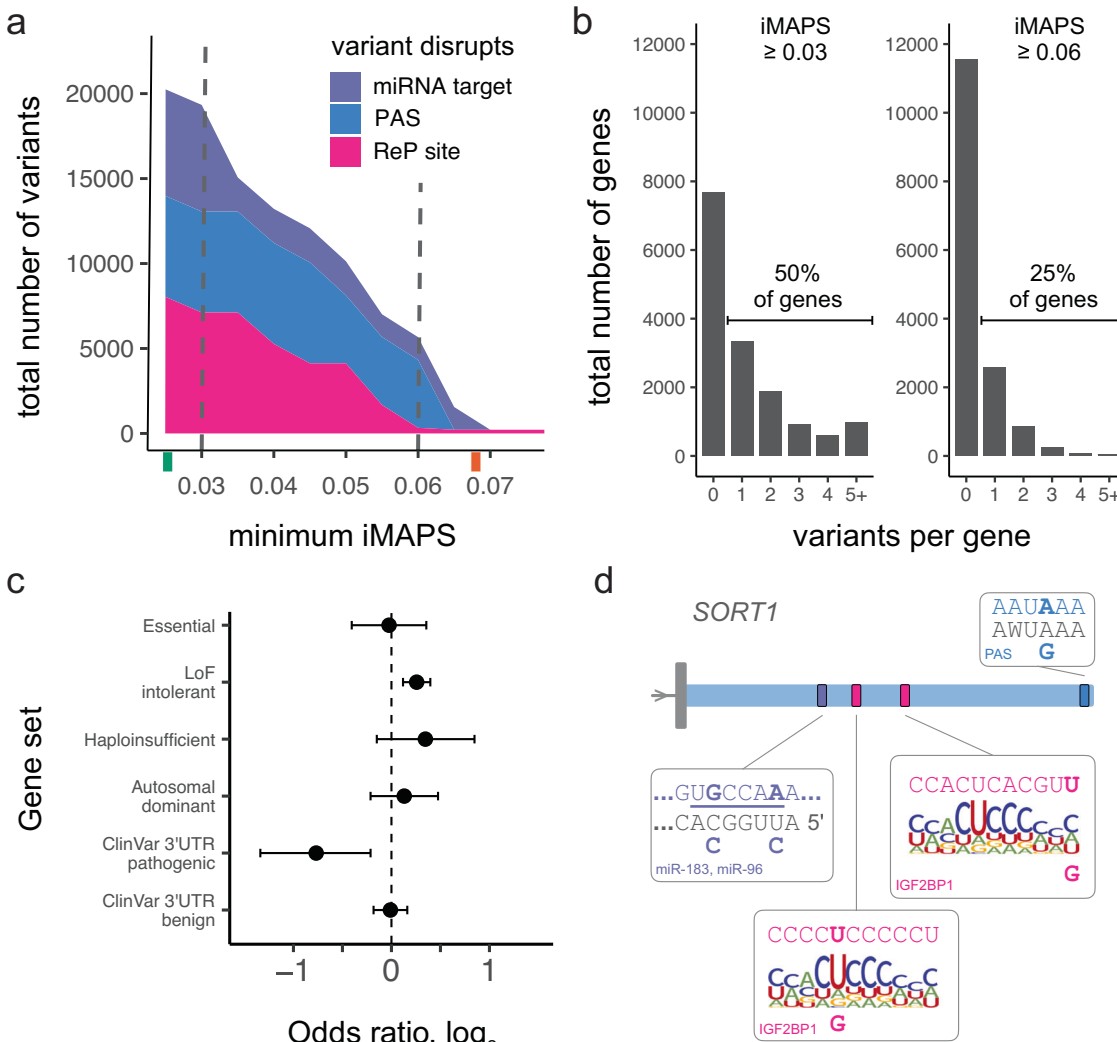

**Fig. 6 | Summary of the number of gnomAD variants belonging to classes under strong negative selection in 3′ UTRs. a** The number of variants disrupting different regulatory elements in 3′ UTRs is shown for categories under increasingly strong selection (minimum iMAPS). Tick marks on the x-axis indicate the genome-wide average iMAPS for synonymous (green) and missense (orange) coding variants. Vertical dashed lines indicate the cut-offs used for gene level analysis in **b**. **b** Histogram of the number of disruptive variants per gene at two different thresholds. Left: half of all protein-coding genes contain at least one 3′ UTR variant labeled "moderately disruptive". Right: one quarter of protein-coding genes contain at least one 3′ UTR variant labeled "highly disruptive". **c** Extent of depletion for

different gene sets among genes with highly disruptive 3′ UTR gnomAD variants. Data are presented as odds ratios with 95% confidence intervals. The dashed vertical line indicates no depletion or enrichment. The number of genes in each gene set are included in Supplementary Data 1. **d** Highly disruptive variants in *SORT1*. For clarity, only the 3′ UTR formed by the primary poly(A) site is shown. For each element, the reference sequence is shown at the top with reference alleles in bold, and alternative alleles are shown at the bottom. Mature miR sequence and top PAS hexamers are shown in gray and the miR seed region is underlined. ReP site variants are shown in the context of an RBPamp affinity model for IGF2BP1.

this summary. In all, one-quarter of protein-coding genes had at least one 3′ UTR gnomAD variant labeled as highly disruptive, and half of protein-coding genes had at least one 3′ UTR gnomAD variant labeled as moderately disruptive (Fig. 6b). Next, as an orthogonal assessment of our classifications, we asked whether the genes with highly disruptive 3′ UTR variants in gnomAD were depleted for types of genes where one might expect disruptive variants to be more deleterious. We uncovered substantial depletion for genes with pathogenic 3′ UTR variants in ClinVar. This suggests that the highly disruptive variants identified are under strong negative selection in genes where 3′ UTR function has been implicated in disease pathogenesis. The highly disruptive 3′ UTR gnomAD variant genes were not, however, depleted for genes with benign 3′ UTR variants in ClinVar, suggesting the observed depletion was specific to evidence of pathogenicity and was unlikely to have resulted from ascertainment bias. Notably, we did not find evidence for the depletion of essential genes or other related gene sets

investigated (Fig. 6C). This may not be surprising given that many of our highly disruptive variants are expected to stabilize transcripts, and that increasing the expression of genes identified by loss-of-function studies may not necessarily be deleterious.

The data in Supplementary Data 2 can be used to augment conventional approaches for 3′ UTR variant interpretation for many human genes. As examples, we highlight two genes with known disease associations and highly disruptive 3′ UTR gnomAD variants: insulin-like growth factor 2 receptor (*IGF2R*) (Supplementary Fig. 8) and sortilin 1 (*SORT1*) (Fig. 6d). *SORT1* encodes a sortilin family receptor involved in intracellular trafficking and is associated with cholesterol levels and myocardial infarction risk[62]. *SORT1* is in the top 7% of loss-of-function-intolerant genes (rank 1,200), in the top 10% of genes by number of variants studied (896)[63], and in the top 5% of genes by number of significant variant-trait GWAS associations (28)[64]. Five *SORT1* 3′ UTR gnomAD variants were labeled as highly disruptive: two disrupting an

8mer target of miR-183 and miR-96, two disrupting distinct IGF2BP1 ReP sites, and one disrupting the PAS for the primary (and conserved) poly(A) site (Fig. 6d). Notably, none of these types of annotations are reported by standard variant interpretation tools such as Variant Effect Predictor (McLaren 2016)[65]. In addition, while some (3/5) of these variants had high phyloP scores, a simple approach to identifying candidate regulatory variants by focusing on positions with phyloP ≥ 2 yielded over 17-fold more *SORT1* 3′ UTR gnomAD variants (a total of 87). As another example, *IGF2R* encodes a receptor for insulin-like growth factor 2 and is often mutated in hepatocellular carcinoma[66]. *IGF2R* is also a loss-of-function-intolerant gene where four 3′ UTR gnomAD variants were labeled as disruptive: three disrupting a PUM2 ReP site, and one disrupting the PAS for the primary (and conserved) poly(A) site (Supplementary Fig. 8). Again, there were over 26-fold more *IGF2R* 3′ UTR gnomAD variants at positions with phyloP ≥ 2 (a total of 107). By comparison, iMAPS-based labels are more specific, sensitive to variant base change and potential human-specific activity, and identify specific regulatory factors.

## Discussion

Recent efforts to catalog human genetic variation using deep sequencing on a genome-wide basis and at the scale of tens of thousands of individuals have enabled study of the signatures of evolutionary selection in our genome[25]. However, it has been difficult to detect signal within non-coding regulatory elements. Here we present the iMAPS method to more accurately and more confidently account for the impact of non-selective forces (such as mutability) on allele frequencies in non-coding regions of the transcriptome. Using iMAPS, we uncovered over 5000 gnomAD variants belonging to classes under strong negative selection. These variants highly disrupt RBP binding, miRNA targeting, or cleavage and polyadenylation. Our approach classified variants using basic principles and data related to these modes of regulation, complementing efforts to interpret 3′ UTR variation using deep learning models[35,67,68]. Layering internal controls for each set of analyses on top of our carefully controlled iMAPS metric lends confidence that the observed effects are driven by selection on variants affecting post-transcriptional gene regulation and not artifacts of forces independent of selection such as mutability, providing a comprehensive and quantitative assessment of negative selection at regulatory elements in human 3′ UTRs.

Many methods for variant interpretation[34,69] incorporate measures of conservation across species, as variants at highly conserved positions are generally more deleterious overall. However, reliance on cross-species conservation to identify functional genetic elements in non-coding regions has significant limitations. First, non-coding regions such as 3′ UTRs are of course less conserved than coding regions[43,70]. Furthermore, there is evidence that non-coding regulatory regions undergo more frequent lineage-specific adaptation, with the extent of lineage-specific constrained sequence rivaling or even exceeding that which is both constrained in humans and conserved across mammals[24,30]. Here, we identify classes of 3′ UTR variants under negative selection in humans without relying on any measures of inter-species conservation, capturing human-specific regulatory constraints even at non-conserved positions. As an example of how constraint can differ within humans and between mammals, we found that variants disrupting PASs associated with less frequently used poly(A) sites in humans were not under strong selection, even in cases where the homologous poly(A) site was utilized in rat and/or mouse (Fig. 5c). A second limitation is that inter-species conservation scores such as phastCons and phyloP do not provide any information as to whether different possible alternative alleles may be differentially tolerated at a given position. We demonstrate the power of our analysis to differentially classify variants based on the consequence of the base change for binding. This is seen most strikingly for RBP binding events where there are large differences in negative selection between disrupting

and preserving variants within the same set of ReP sites, even when restricted to variants at the most highly conserved positions (Fig. 2g). Conventional one-dimensional approaches to variant interpretation typically prioritize all variants within an annotation. We show evidence that utilizing multi-dimensional annotations (e.g., ReP sites derived using both eCLIP and RBPamp data) is critical to uncovering classes of variation under strong selection, and that an approach sensitive to the impact of different potential base changes introduced by genetic variants can filter more neutral classes of variants within these annotations.

Our work uses orthogonal human population genetic data to validate the utility of several RNA related datasets for detecting regulatory elements in 3′ UTRs. We find that the majority of selection against PAS-disruption, miRNA-disruption, and Pumilio binding site creation occurs in core 3′ UTR regions upstream of the most utilized poly(A) site in a gene, based on average utilization across various cell lines and tissues from polyA_DB[47], suggesting that this consideration is extremely useful in guiding the search for functional elements in 3′ UTRs. Based on the notion that variants within expressed mRNA isoforms are more likely to be functional, we provide evidence that the APA context needs to be explicitly considered when interpreting 3′ UTR variants, as has proven useful in identifying miRNA targets[71]. These results emphasize the utility of a recently developed "expression-aware" variant interpretation framework that considered how often variants were found within alternatively spliced transcript regions[72], and argue for similar practices to be applied in 3′ UTRs based on APA. We also demonstrate the utility of performing eCLIP across multiple cell lines, as RBP-disrupting variants within ReP sites supported by eCLIP peaks across two different cell lines are associated with stronger negative selection.

We also find that RBPamp affinity models[45] derived from in vitro RBNS data[38,73] are insufficient by themselves to identify constrained RBP binding sites (Fig. 2e). However, when complemented by in-cell eCLIP data[20], these affinity models prove crucial for the identification of the precise binding sites associated with eCLIP peaks that are under selection. In describing these so-called ReP sites, we provide a powerful set of precise and high-confidence binding sites for a diverse set of RBPs. In addition to demonstrating that these sites are under negative selection, we also show that they are enriched for regulatory function. ReP sites fill an important gap in variant interpretation efforts in the context of general RBP binding. Moreover, at ReP sites, RBPamp demonstrates utility in suggesting a general minimum relative affinity that constitutes functional binding, and in distinguishing between variants that disrupt or preserve binding.

While using population genetic data from contemporary human populations cannot assess negative selection acting on an individual genetic variant, our results can inform variant interpretation. The stratification of variants performed here provide useful thresholds and benchmarks for interpretation of both known and not yet observed variants that may impact different post-transcriptional gene regulatory mechanisms, enabling more confident interpretation of disease-associated non-coding variants (Fig. 6 and ref. 74). Specifically, we were able to label between 5,000 and 20,000 gnomAD variants in 3′ UTRs as moderately or highly disruptive, respectively, and highlight examples of haploinsufficient disease-associated genes with several highly disruptive 3′ UTR variants. While gnomAD is a useful reference, these numbers can be thought of as minima, as consideration of additional rare variants from other sources increases the number of labeled variants (Supplementary Data 2). Overall, our findings deliver detailed insight into the nature of RBP activity in untranslated regions, helping us to continue to discover how these fundamental interactions regulate gene expression in ways that have and will continue to shape human biology.

## Methods

### Genetic variants

Genetic variants from gnomAD release 3.0 were downloaded from https://gnomad.broadinstitute.org/downloads in vcf format. Single nucleotide variants passing gnomAD quality filters were isolated using VCFtools[75]. Variants observed only once in the gnomAD dataset were considered "singletons". Genetic variants from denovo db (from non-SSC samples) were downloaded from https://denovo-db.gs.washington.edu/denovo-db/Download.jsp.

### Variant filtering for negative selection analyses

To reduce bias in the allele frequency spectra that might confound negative selection analyses, we filtered variants in: low complexity regions (defined by gnomAD), sex and mitochondrial chromosomes (X, Y, and M), regions with median sequencing coverage less than 25 (3/4 or 75% of the genome-wide median coverage of 32) or greater than 42 (4/3 or 133% of the genome-wide median coverage of 32), the 'C' position of CpG dinucleotides without any available methylation data, and CpG islands[76,77]. To ensure we were studying the impact of the newly derived allele, we excluded variants where the reference allele was not also identified as the ancestral allele using the *Homo sapiens* ancestor genome sequence from Ensembl (https://ftp.ensembl.org/pub/current_fasta/ancestral_alleles/homo_sapiens_ancestor_GRCh38.tar.gz). To limit the impact of transcription-associated mutability, all variants with base change of A > G on either strand (i.e., A > G or T > C) were excluded. Variants with any "N" base calls for the five nucleotides on either side of the variant were also excluded. Multi-allelic sites were retained and treated as independent variants. Variants in 3′ UTRs that were also within any Gencode CDS exons (on either strand) were excluded from analysis.

### Variant annotations

Variant positions were then annotated as either overlapping or non-overlapping with DNase hypersensitivity peaks (hg38wgEncodeRegDnaseClustered from the UCSC Table Browser), CpG islands (hg38cpgIslandExtUnmasked from the UCSC Table Browser), and H3K9me3 peaks (E062-H3K9me3.bed downloaded using https://github.com/carjed/smaug-genetics/blob/master/download_ref_data.sh). CpG transition variants were intersected with methylation data (processed data from Roadmap Epigenomics Consortium 2015; provided by gnomAD). These variants were assigned a methylation level as in[25]: High = mean methylation > 60, medium = mean methylation ≤ 60 and > 20, low = mean methylation ≤ 20. Variants at positions with missing methylation data were not assigned a bin and were later excluded.

### Intergenic regions

The intergenic regions used to derive the expected proportion singleton values for calibration of iMAPS scores were selected as follows: All regions at least 25 kb from any Gencode v32 entry were eligible. From these regions, gap locations and centromere regions were then excluded. To avoid disproportionate contributions from very large intergenic stretches in the genome, long continuous regions were trimmed to a maximum length of 1.5-times the 75th percentile length. This resulted in a set of intergenic regions totaling approximately 330 Mb.

### Quantifying negative selection

Both iMAPS and MAPS estimate negative selection for a given set of variants as the rate of excess singletons (variants observed only once in gnomAD), using Eq. 1, where the "expected singletons" value is the sum of the expected proportion singleton values assigned to each variant:

$$\text{(i)MAPS} = \frac{\text{observed singletons} - \text{expected singletons}}{\text{number of variants}} \quad (1)$$

The two approaches differ in how they derive these expected proportion singleton values to calibrate variants of interest (and (i) MAPS scores) to a more neutral reference region of the genome:

1. The MAPS approach derived expected proportion singleton values for genomic contexts that considered all possible combinations of base change + dinucleotide context + methylation level (for C to T transitions at CpG loci). For each context, relative mutation rates were estimated using variants in intergenic and intronic regions, which were then "converted" to expected proportion singleton values by regressing observed proportion singleton values from synonymous CDS variants against these relative mutation rates. Each variant of interest was then assigned an expected proportion singleton value (which can theoretically vary between 0 and 1) corresponding to the proportion singleton value returned by the regression for the matching context[31].

2. In contrast, the iMAPS approach introduced here derived expected singleton values for genomic contexts that considered all possible combination of base change + (up to) hexa-nucleotide context + methylation level (for C to T transitions at CpG loci) + DNase hypersensitivity peak overlap (binary) + H3K9me3 peak overlap (binary). For each context, proportion singleton values were obtained using variants in the select intergenic regions described above. Each variant of interest was then assigned an expected proportion singleton value (which can theoretically vary between 0 and 1) corresponding to the intergenic proportion singleton value for the matching context. Additionally, the iMAPS approach also accounted for known influences of CpG islands[29] and transcription[39] by filtering variants in CpG islands and those with an A > G base change, respectively. Notably, iMAPS does not rely on CDS variants for calibration. Collectively, iMAPS more extensively accounts for non-selective factors influencing allele frequency spectra (as summarized by proportion singleton values), which resulted in improved calibration relative to MAPS (Supplementary Fig. 1). An iMAPS calculator that uses this approach to quantify negative selection acting on user-provided 3′ UTR elements or variants is available at https://github.com/sfindlay11/iMAPS.git. To test for significant differences in negative selection between a group of variants we hypothesized to be under strong selection and a control group of variants, we performed one-sided Fisher exact tests comparing the number of observed singletons and non-singletons between the groups. Importantly, we tested against an odds ratio derived from comparing the number of expected singletons and non-singletons derived from the matched intergenic variants. Exact *P* values are reported for figure panels with a single statistical test. For those with more than one statistical test (for example testing at different minimum affinity thresholds), we used asterisks (*) to indicate tests with $P < 0.05$.

### Accounting for the impact of extended flanking sequence contexts

To account for the impact of extended flanking sequence contexts on allele frequency spectra, a set of 416 "parent contexts" was first generated to describe all possible local genomic contexts at the dinucleotide level as follows: 43 = 64 dinucleotide contexts (including the 4 NCG > NTG contexts) x 3 base changes = 192 reverse complement pairs = 96 distinct contexts + 8 for NCG > NTG at two additional methylation levels = 104 x 2 (for +/− DNaseHS peak binary) = 208 x 2 (for +/− H3K9me3 peak binary) = 416 parent contexts. An example parent context is "C[A > G]G, DNaseHS-positive, H3K9me3-negative." Given that previous literature has established that nucleotide contexts beyond dinucleotide frequently impact mutability (e.g. (Carlson 2018)), and that the abundance of intergenic variants provides adequate statistical power, we incorporated extended nucleotide context into our iMAPS metric by accounting for contexts up to

hexanucleotide (5 bases on each side of the variant). Since simply utilizing the proportion singleton for every possible hexanucleotide context ($4^{11}\cdot3 = 65{,}536$ additional contexts for each of the 416 parent contexts) had the potential to introduce a lot of noise for infrequently observed contexts, we took a more conservative approach: Given that dinucleotide context offered the most statistical power, we used the proportion singleton value observed for intergenic variants for each parent context as a baseline value when assessing more extended contexts. Next, we compared each hexanucleotide context to its parent dinucleotide context using two-sided Fisher exact tests and Benjamini-Hochberg correction for multiple hypothesis testing to determine if the extended context had a significantly different proportion singleton than its parent context at an FDR of 0.2. If it did, we used the proportion singleton value from the extended hexanucleotide context for variants with this context. If it did not, we repeated this process iteratively for the embedded penta-, tetra-, and tri-nucleotide contexts, respectively, defaulting to the dinucleotide parent context if no extended contexts were found to significantly influence the proportion singleton values.

### Coding sequence (CDS)
Gencode v32 CDS exons were downloaded from the UCSC Table Browser. Only CDS exons with transcript ids matching Gencode v32 protein-coding transcripts from protein-coding genes were analyzed. Gencode phase information was used to determine codon positioning and protein-coding consequence. CDS exons with any amount of overlap with another CDS exon annotation with a different reading frame were excluded from analysis. Variants within three bases of any splice site were excluded from analysis. A very small number of variants where the ancestral allele coded for a stop codon were excluded from analysis.

### Human 3′ UTRs
Human 3′ UTRs were defined using polyA_DB version 3.2[47]. Each poly(A) site was matched to the closest upstream stop codon annotated in Gencode v32 to form a 3′ UTR. Only poly(A) sites downstream of annotated stop codons from protein-coding genes were considered for analysis. Resulting 3′ UTRs longer than 50 kb were filtered from analysis. For identification of ReP sites within 3′ UTRs, Gencode v32 3′ UTRs were also used for genes with no poly(A) site data in polyA_DB. For all analyses where core and variable 3′ UTR regions were stratified (see below), only polyA_DB inferred 3′ UTRs were used.

### RBPamp
Position-specific affinity matrices (PSAMs) of length $k = 10$ or $k = 11$ generated by RBPamp[45] were used to score the affinity of a given RBP for potential target RNA sequences relative to the RBP's ideal binding site (affinity = 1.0). We used PSAMs that were generated by considering both the sequence and predicted secondary structure of random sequence target RNA oligos in RBNS experiments.

### RBPamp eCLIP-proximal (ReP) sites
eCLIP peak data for K562 and HepG2 cell lines were downloaded from the ENCODE data portal. Peaks that were reproducible (IDR) across biological replicates were used for analysis. RBPamp PSAMs were used to score the relative affinity of every possible binding site within 75 bases of the 5′ end of each eCLIP peak (in either direction). Reference sequence was used as input. We identified the highest affinity site associated with each eCLIP peak. We termed these sites "RBPamp eCLIP-proximal" or "ReP" sites and considered the union of ReP sites across both cell lines. Since eCLIP peaks are typically > 50 bases (and often hundreds of bases) in length, and the RBPamp models consider a maximum binding footprint of $k = 10$ or $k = 11$ bases (depending on the RBP), ReP sites are a more precise set of high-confidence RBP binding sites. For most RBP-cell line pairs, the highest scoring sites were highly

enriched at or close to the 5′ end of eCLIP peaks. This is expected from eCLIP experiments, given that cross-linked RNA bases often interfere with reverse transcription, causing read pileup and subsequent peak calling at or downstream (3′) of cross-linking sites. Identification of the highest affinity sites was also used to validate RBPs for which eCLIP and RBPamp data were coherent: For the union of all peaks across cell lines for each RBP, we determine how frequently each position (+/− 75 bases relative to the 5′ end of each peak) overlapped the highest affinity site. We then conducted a simulation where we select a random site for each peak (+/− 75 bases from the 5′ end of the peak) with equal probability across all positions. We repeat this 10,000 times to obtain a *P* value of how often the simulated data results in a single position that is observed as frequently as or more frequently than the most frequently observed position for the real set of highest affinity sites. ReP sites for RBPs with *P* values < 0.01 were retained for negative selection analyses. This resulted in filtering of only a few RBPs, most with very few eCLIP peaks in 3′ UTRs (Supplementary Fig. 2).

Control sites used in Fig. 2e were generated by first masking all ReP sites so any site overlapping any part of a ReP site would receive an affinity score of 0. For each ReP site, we then identified the site in the same 3′ UTR with the closest affinity (either higher or lower). Variants in these sites were classified using the same method applied to variants in ReP sites as described below.

For RBPs with eCLIP data available for both K562 and HepG2 cells, ReP sites were considered shared across cell lines in Fig. 2f if the exact same ReP site was identified in each cell line for the same RBP, regardless of whether the associated eCLIP peaks matched exactly. Partially overlapping but not completely matching ReP sites were not considered shared across cell lines. Since ReP sites shared across cell lines required an eCLIP peak to be called twice (once in each cell line), these ReP sites may be enriched for highly expressed genes (that are more constrained in general) relative to ReP sites identified in only a single cell line. Therefore, we used a control set of ReP sites from genes where 3′ UTR ReP sites were detected for the same RBP in both cell lines, but at different positions (i.e. they were not the exact same ReP site).

### MPRA transcript abundance modulating variants (tamVars)
TamVar data was obtained from[46]. Variants with statistically significant tamVar activity in any cell line tested were considered significant. We tested the odds ratio (odds of ReP site variants having significant tamVar activity in the MPRA relative to all other variants tested) at decreasing *P* value cut-offs (increasing stringency).

### ReP site conservation across species
For each ReP site (10 or 11 bases in length), a control eCLIP peak region of the same length within (or closely upstream of the 5′ end of) the peak was selected. For each RBP, the positions of the eCLIP control regions were assigned by randomly sampling from ReP site positions (relative to the 5′ ends of their corresponding eCLIP peaks). This was to ensure that the control eCLIP regions had positional distributions similar to the ReP sites to which they were being compared. Any candidate control regions that overlapped the matching ReP site at all were passed over and sampling was repeated until a non-overlapping region was assigned. ReP site-control region pairs with any CDS overlap were excluded from analysis. phyloP (100-way) scores were downloaded from http://hgdownload.cse.ucsc.edu/goldenPath/hg38/phyloP100way/ and intersected with ReP sites and control regions. All positions within any ReP sites above the minimum affinity threshold and its matched control region were analyzed collectively. Bases were considered conserved if their phyloP score was above a varying threshold between 2 and 6. The odds ratios were calculated as: $(\text{Conserved}_{\text{ReP}} / \text{Non-conserved}_{\text{ReP}}) / (\text{Conserved}_{\text{match}} / \text{Non-conserved}_{\text{match}})$ and 95% confidence intervals were calculated using Fisher exact tests.

## Classification of ReP site variants

ReP site variants were classified as either focal or non-focal and disrupting or preserving. First, for each variant allele, we summed the relative RBPamp affinities across all positions overlapping the variant (i.e., 11 affinities for a width k = 11 RBPamp model) and termed these the ancestral affinity and the derived affinity. Next, we summed the relative affinity values spanning from 25 bases upstream to 25 bases downstream of the ReP site and termed this the local affinity. Focal variants were those with ancestral affinity / local affinity > 2/3 (i.e., the majority of the local affinity could be attributed to the ReP site). Non-focal variants were those with ancestral affinity / local affinity <1/3, or with another ReP site for the same RBP within 25 bases. Disrupting variants were those with derived affinity / ancestral affinity <1/3, and preserving variants were those with derived affinity / ancestral affinity > 2/3. Negative selection analysis of disrupting and preserving variants was performed on variants aggregated across all RBPs with available data (after filtering as described above). Analysis of individual RBPs was not possible due to low statistical power.

## eQTLs

DAG-P fine-mapped eQTL variant call files were downloaded from the GTex Portal. For each variant, the maximum posterior inclusion probability (PIP) value across tissues was used. Due to the relatively small number of ReP site variants with fine-mapped eQTL data available, we used more relaxed cut-offs for defining disrupting (derived affinity / ancestral affinity <1/2) and preserving variants (derived affinity / ancestral affinity > 1/2) for the analysis in Fig. 2d. Notably, it is not possible to analyze negative selection acting on variants from association-based analyses such as GWAS and eQTL, as they are highly skewed toward common variants (with more statistical power to detect significant effects) and measures such as iMAPS depend on unbiased allele frequencies.

## Parallel reporter assay

To test variants for their ability to directly modulate transcript levels in cells, a parallel reporter assay was conducted. 28 pairs of 101 nucleotide 3′ UTR fragments consisting of two variants alleles +/− 50 bases of flanking sequence were ordered in oPools Oligo Pool format (IDT) and cloned into a modified version of the pmirGLO (Promega) plasmid (Griesemer 2021) downstream of a GFP ORF via BsaI sites introduced by site-directed mutagenesis (Agilent). The plasmid library was transfected into HEK 293 cells using Lipofectamine 3000 (Life Technologies) and total RNA was collected 48 hours later using the RNeasy mini kit (Qiagen). Plasmid RNA was reverse-transcribed using a gene-specific primer (GCATTCTAGTTGTGGTTTGTCCA) and Superscript IV (Life Technologies). Libraries were amplified and each sample (two replicates of input plasmid DNA and two RNA samples from separate transfections) was uniquely dual indexed using custom primers. RNA and plasmid DNA input read counts for each fragment were obtained using Illumina Miseq performed by the BMC at MIT. For each variant, the variant activity is the odds ratio calculated as: $(RNA_{Alt} / DNA_{Alt}) / (RNA_{Ref} / DNA_{Ref})$, where Alt = alternative allele and Ref = reference allele. Variants with Benjamini-Hochberg corrected Fisher exact test $P < 0.05$ for both transfections were considered to have significant capacity to modulate transcript levels. HEK 293 cells were cultured in DMEM (Life Technologies) supplemented with 10% FBS (Life Technologies). Negative control variants were selected from variants assayed in[46] with <10% skew in HEK 293 cells, and oligo pairs containing the two nonassayed alleles (e.g. U and C for an A > G variant on the sense strand) were synthesized and tested. Two variants where both of the nonassayed alleles have both been observed (present in dbSNP) were excluded from analysis. Plasmid and RNA read counts for ReP site-disrupting and negative control variants are provided in Supplementary Data 3.

## Highly disruptive 3′ UTR gnomAD variant gene set analyses

Genes with highly disruptive 3′ UTR gnomAD variants belonging to classifications with iMAPS >= 0.06 were analyzed. For each class of disrupting variants (ReP, PAS, and miR) an equal number of control genes were sampled from genes without any moderately disruptive or highly disruptive 3′ UTR gnomAD variants. For Rep sites, highly disruptive variants consisted of those that disrupted ReP sites with relative affinities > 0.01. Eligible control genes had at least one variant in a ReP site with a relative affinity of > 0.01. For miR targets, highly disruptive variants were found in core 3′ UTR regions within 8-mer target sites for more than one broadly conserved microRNA family. Eligible control genes had at least one variant in their core 3′ UTR region within an 8-mer miR target. For PASs, highly disruptive variants caused loss of a PAS within 15 to 30 bases of a gene's primary poly(A) site that was also used in mouse or rat. Eligible control genes had at least one variant upstream of a primary poly(A) site that was also used in mouse or rat. Gene lists for various annotations were downloaded from https://github.com/QingboWang/gene_lists: Essential genes were identified in a CRISPR-based cell viability screen[78]. Haploinsufficient genes were identified from ClinGen[79]. Autosomal dominant genes followed an autosomal dominant inheritance pattern[80].

## Expression data

Gene expression summary data for HepG2 and K562 cell lines were downloaded from the ENCODE Portal. Genes with ids matching Gencode v32 protein-coding genes were analyzed. Mean transcripts per million values across replicates were used.

## Defining core and variable 3′ UTR regions

We used polyA_DB version 3.2 data to identify core and variable 3′ UTR regions[47]. These regions were defined based on relative utilization of different poly(A) sites, not necessarily their relative position. We defined the primary poly(A) site for each gene as the site with the highest mean reads per million across all polyA_DB tissues and cell lines. Thus, our primary poly(A) sites capture the effects of both alternative last exon selection and alternative polyadenylation. All positions between the primary poly(A) site and its upstream stop codon, regardless of any intervening poly(A) sites, were considered core. All positions between the primary poly(A) site and the most distal poly(A) site were considered variable. If the most distal poly(A) site associated with a given stop codon was also the primary poly(A) site, we considered that 3′ UTR to have no variable region.

## Pumilio site creation

We used the RBPamp model for PUM1/PUM2 to predict the relative ancestral and derived affinities for all 3′ UTR gnomAD variants as described above. RBPamp was used to score relative affinity for PUM1/PUM2. Variants with derived affinity / ancestral affinity > 2 were included for negative selection analysis in Fig. 3b. All variants with alternative affinity > reference affinity were included for analysis of mean eQTL PIP. For eQTL gene expression analysis, the normalized effect size (NES) values obtained from GTEx represent the relative expression difference between transcripts containing the alternate allele and transcripts containing the reference allele, where positive values indicate higher expression for the alternate allele, and vice versa. eQTL variants with alternative affinity > 0.3 and alternative affinity > reference affinity were labeled as "created." Variants labeled as "none" consisted of all other variants after excluding variants with reference affinity > 0.3 and those in PUM2 ReP sites. A Wilcoxon Rank Sum test was performed to compare NES values between the "none" and "created" groups. Each PIP-tissue combination was considered for each eQTL variant.

## miRNA target sites

All miRNA target site data was downloaded from https://www.targetscan.org/vert_80/vert_80_data_download/All_Target_Locations.hg19.bed.zip. Only 8mer, 7mer-m8, and 7mer-A1 target sites (including both conserved and non-conserved sites) of miRNA families conserved in mammals or vertebrates were considered, and targets of poorly conserved miRNA families were excluded. All considered targets were grouped together for analysis of miRNA targets in Fig. 1d. For all analyses variants in the following target positions were analyzed: bases within the seed pairing with positions 2 through 7 of the miRNA (all types), the base across from position 1 (for 8mer and 7mer-A1 types), and the base pairing with position 8 (8mer and 7mer-m8 types).

## Polyadenylation signal (PAS) analysis

"Top" PAS hexamers (with ancestral sequence AWUAAA, where W = A or U) 15 to 30 bases (inclusive) upstream of primary poly(A) sites identified from polyA_DB data (see above) were considered primary PASs. All other PASs (including those nearby any nonprimary poly(A) sites in core regions) were considered secondary. PASs were considered lost if the derived allele sequence did not overlap any PAS hexamer, including both AWUAAA and other similar A-rich hexamers found to be enriched in PASs from[58]. Variants with derived allele sequence that overlapped one or more PAS hexamers were considered preserved. Conserved and nonconserved (active in human and either rat or mouse) poly(A) site classifications were obtained from polyA_DB[47]. To account for poly(A) sites that were difficult to confidently assign to a single stop codon/3′ UTR in the absence of long-read sequencing data, we filtered pairs of 3′ UTRs where any poly(A) sites are within 100 bases of a downstream stop codon / 3′ UTR start.

## Comparing iMAPS across modes of regulation

We scaled ReP site iMAPS values to facilitate comparison between all three modes of regulation investigated. ReP sites are derived from eCLIP peaks, which are enriched in highly expressed genes where higher read counts increase the statistical power for peak calling. Since highly expressed genes are more constrained, we expect genes containing ReP sites to have slightly elevated iMAPS relative to genome-wide averages. In contrast, miRNA targets were predicted and analyzed for all genes, and PASs were considered equally across all transcripts with a detectable 3′ end / poly(A) site.

We calculated iMAPS for synonymous and missense variants from genes with 3′ UTR ReP sites to scale ReP site variant iMAPS values. Specifically, we sampled CDS variants from genes with 3′ UTR ReP sites in a weighted fashion to reflect the number of ReP sites in each gene. These CDS variants were then classified and iMAPS was calculated as described above, resulting in iMAPS of ~ 0.034 and 0.10 for synonymous and missense variants, respectively. These values were used as relative benchmarks to convert ReP site iMAPS values to the genome-wide scale with synonymous and missense iMAPS of 0.028 and 0.070, respectively. For example, a ReP site iMAPS value of 0.067 (half-way between synonymous and missense ReP site gene values) would scale to 0.049 (half-way between synonymous and missense genome-wide values).

After scaling of ReP site iMAPS values as described above, we counted the number of gnomAD variants belonging to any 3′ UTR element-disrupting classification with iMAPS values above a minimum threshold ranging from 0.025 to 0.075 (in increments of 0.005). We chose 0.03 and 0.06 as minimum iMAPS thresholds to highlight as they exceeded genome-wide synonymous levels and approached genome-wide missense variant levels, respectively. None of the filters described in "Variant filtering for negative selection analyses" above were applied when counting the number of gnomAD variants belonging to each classification.

## Data analysis software

Standard computational biology software, including R (v4.2.0), Python (v3.8.5), bedtools (v2.29.1), and vcftools (v0.1.17) was used in data analysis. Details for RBPamp and custom software can be found in the "Data availability" and "Code availability" sections, respectively.

## Reporting summary

Further information on research design is available in the Nature Portfolio Reporting Summary linked to this article.

## Data availability

Several publicly available datasets were utilized in this study and can be accessed as follows: gnomAD variants; https://gnomad.broadinstitute.org/downloads denovo db variants; https://denovo-db.gs.washington.edu/denovo-db/Download.jsp Ancestral human genome sequence; Ensembl: https://ftp.ensembl.org/pub/current_fasta/ancestral_alleles/homo_sapiens_ancestor_GRCh38.tar.gz DNase hypersensitivity peaks and CpG islands; UCSC Table Browser: https://genome.ucsc.edu/cgi-bin/hgTables Gencode gene regions; UCSC Table Browser: https://genome.ucsc.edu/cgi-bin/hgTables, and Gencode: https://www.gencodegenes.org/human/release_32.html H3K9me3 peaks; accessed using https://github.com/carjed/smaug-genetics/blob/master/download_ref_data.sh CpG methylation data processed by the Roadmap Epigenomics Consortium can be requested from gnomAD polA_DB data; https://exon.apps.wistar.org/PolyA_DB/v3/misc/download.php RBPamp; https://bitbucket.org/marjens/rbpamp ENCODE eCLIP and gene expression data; https://www.encodeproject.org/ Processed tamvar MPRA data; https://www.cell.com/cms/10.1016/j.cell.2021.08.025/attachment/d8a42d2a-9add-4815-b0ce-f4e1b28d5ca9/mmc1.xlsx phyloP scores; http://hgdownload.cse.ucsc.edu/goldenPath/hg38/phyloP100way/ eQTLs; https://www.gtexportal.org/home/ Annotated gene sets; https://github.com/QingboWang/gene_lists ClinVar variants; UCSC Table Browser: https://genome.ucsc.edu/cgi-bin/hgTables miRNA targets; https://www.targetscan.org/vert_80/vert_80_data_download/All_Target_Locations.hg19.bed.zip Reporter assay RNA and plasmid read counts can be found in Supplementary Data 3.

## Code availability

Custom code, along with an iMAPS calculator that uses our approach to quantify negative selection acting on user-provided 3′UTR elements or variants is available at https://github.com/sfindlay11/iMAPS.git.

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

## Acknowledgements

We thank members of the Burge laboratory for their helpful discussions and comments on the manuscript, especially Marvin Jens and Hannah Jacobs for their insights. We thank Nicola Whiffin for helpful discussions and Konrad Karczewski for facilitating access to useful datasets. We thank James Xue and Dustin Griesemer for guidance on parallel reporter assays. We thank the staff of the BMC at MIT for their sequencing services and technical support. We also thank David Bartel and Evan Boyle for providing helpful comments on the manuscript. S.D.F. was supported by a postdoctoral fellowship from the Natural Sciences and Engineering Research Council of Canada (NSERC). This work was funded by grants from the NIH (GM085319 and HG002439 to C.B.B.).

## Author contributions

S.D.F. designed the study with input from C.B.B.; S.D.F. performed analysis with contributions from L.R.; S.D.F. wrote the draft manuscript; S.D.F. and C.B.B. finalized the manuscript with input from L.R.

## Competing interests

The authors declare no competing interest.
