## [Peer Review File · Nature Communications]

Quantifying negative selection in human 3' UTRs uncovers constrained targets of RNA-binding proteinsREVIEWER COMMENTS

Reviewer #1 (Remarks to the Author):

This manuscript describes a modification to the established MAPS (Mutability Adjusted Proportion Singleton) method, specifically designed to calibrate to the mutagenic propensities of 3' UTR, which may be different from those in coding regions. This is an important problem, as the functional significance of variants in 3' UTR are, without doubt, currently under-appreciated. The original MAPS corrects for the confounding effects of differential mutagenicity when assessing the evidence for negative selection. The authors introduce iMAPS, which performs the calibration based on intergenic regions broadly, though the method was applied here specifically to the interpretation of variants in 3' UTRs of genes. The authors appeared to also have stratified by varying amounts of sequence context, which seems appropriate. One question I have is whether the calibration could be improved by focusing specifically on pooled UTR rather than admitting all intergenic stretches, given that many gene loci are transcribed at substantially higher levels and may have different mutagenicity profiles than so-called intergenic "deserts".

The authors provide extensive analyses and make a strong case that their method uncovers clear signs of selection on known functional elements such as RNA-binding-protein binding sites, miRNA target sites, and polyadenylation signals, and observe different selection on constitutively versus alternatively included locations after cleavage and polyadenylation, consistent with recent results on the use of alternative polyadenylation sites in humans and their probable adaptive role. These nuanced analyses appear to follow quite well our current understanding of the biology of post-transcriptional gene regulation through mechanisms involving 3' UTR. Moreover, the authors demonstrate that their method is more specific than broad conservation signals quantified by PhastCons/PhyloP, which do not (though technically they could) parse their signals down to the level of individual nucleotide substitutions. They also make the important point that the evidence for recent adaptation in regulatory regions on the human lineage suggests alternative analytic approaches such as this one.

I do think this manuscript could benefit from a better description of the method for non-specialists (including computational non-specialists). In particular, I think it would be useful to more directly contrast differences in the mathematical formulation of the iMAPS and MAPS methods. My understanding is that the original MAPS was a regression model, whereas this appears to be a contingency table-based method.

In summary, I think the problem addressed by the authors is potentially an extremely important one, given the importance of accurate variant interpretation in the context of disease studies and patient diagnosis, and the generally under-appreciated nature of gene regulation via 3' UTR. Extensive analyses by the authors convincingly argue that these types of modifications to existing approaches to variant interpretation in non-coding regions deserve attention.

Reviewer #2 (Remarks to the Author):

Authors introduced a new method to calculate the gene expression-related impact of non-coding variants in post-transcriptional level, mostly focused on negative selections of the variants by considering population-level germline variations. The method was named as iMAPS (intergenic mutability-adjusted proportion singleton), an improved method of the MAPS, with calibrating of observed singleton variant numbers with expected number of singleton variants using extended flanking k-mers. The major advances of this method compared to the previous method appear to be that 1) removal of singleton variants that associated with transcription to clearly capture the post-transcriptional signals and; 2) utilized orthogonal approaches to prioritize regulatory motifs in 3' UTR ;

3) used intergenic region as control and 4) larger sequence context to calculate expected number of singleton variants (upto five in a side); 5) utilized the GnomAD database to estimate iMAPS of variants from 3'UTR contexts. With a naïve analysis, all variants in 3'UTR displayed lower iMAPS in eCLIP-RBP sites, miRNA sites, polyA signal, than synonymous mutations, with showing less negative selection pressure in 3'UTR than in coding regions. However, prioritizing the variants using orthogonal datasets (BIND-N-Seq. and conservation), they found that some variants sets displayed greater negative selection than synonymous mutations, suggesting that the iMAPS method could unprecedentedly capture functional non-coding variants in 3'UTRs. Although the authors presented interesting and unprecedented reports, there are some descriptions and definitions that they have to clarify in interpreting the results as well as there are technical issues that they should be carefully addressed.

[Major comments]

1. Authors claimed that the iMAPS detected functional non-coding variants in 3'UTRs using orthogonal datasets and information, some of which displayed greater negative selection than synonymous mutations, however previous methods, such as MAPS and methods prioritizing variants on conservation scores, could still capture those same functional variants with lower signal though. I wonder how the scores are correlated among iMAPS and other previous methods.
2. Related to comment 1, I wonder if the iMAPS outperformed previous methods in terms of sensitivity and specificity. They need to carefully demonstrate the improvement of the suggested method using standard data to see whether they capture more true positives and less false positives.
3. I wonder the rationale of the use of intergenic region and larger sequence context for calculating expected number of singleton variants. Authors claimed the power of the use of them with the analysis of functional context of 3'UTR, but they failed to clearly show the improvement over conventional method when using authors' method. Rather they just prioritized functional variants using orthogonal datasets.
4. Moreover, they selected different number of flanking sequence context by each functional group of 3'UTR. However, it can make confound. For example, if we want to compare two sets where the used flanking sequence lengths are measured differently, it may produce different results than when comparing them separately. Further clarification on methodological justification is needed.
5. Synonymous mutations show their phenotypes through changes in mRNA stability, RNA structure, translation kinetics, and splicing. The negative selection for the synonymous mutations would seem to be varied (splicing could be greater than others), so if authors analyze synonymous mutations separately by the biological consequences, they could see different results.
6. Throughout the study, they mostly tried to capture the computational association and enrichments to find functional non-coding variants with no experimental validation. If they deliver an experimental validation for core findings to show the functional non-coding variants are actual, it could strengthen their works further.
7. In some analysis, they focused on disrupting motifs but others focusing on the creation and strengthened motifs to prioritize functional variants without clear rationales. They need to clarify why they focused on one side.
8. There are many RBPs in the analysis of figure 2C. Although the authors showed the negative effect of creation of PUM binding sites on the transcript levels in figure 3C, it would be very helpful to understand each protein function if the authors show the respective effect of RBP binding site deletion of each RBP on the transcript level, besides eQTL PIP changes.
9. ReP sites seem to be more relevant to capture the functional non-coding variants in 3'UTRs. I wonder if the ReP sites are more evolutionally conserved than eCLIP sites and non-ReP sites in eCLIP sites, and wonder if the disrupting ReP sites are less/more conserved than preserving sites. In addition, I wonder if the ReP sites are more negatively selected in core regions of 3'UTRs than in variable regions.
10. They found some association between negative selection and AU flanking of canonical miRNA 8-mer sites. I wonder if they also found the relationship within the sites along with flanking regions.

[Minor]

1. Authors used different number of flanking sequences to calculate expected number of singleton

variants according to variant group, but it is not indicated in paper. Authors should indicate these numbers.

2. Clarify the intended meaning of "calibration" clearly used in page 5 and throughout the manuscript.

3. For general reader, they need to explain what is MAPS in the abstract briefly because they used iMAPS term first in the abstract.

Reviewer #3 (Remarks to the Author):

Summary of content

Findlay, Romo, and Burge develop a measure of non-coding region negative selection and report negative selection comparable to missense variants for groups of 3'UTR variants at specific sites (RBP binding sites, miRNA target sites, and polyadenylation signals). Overall, in a series of well done analyses calculating the negative selection metric "iMAPS", the study confirms the functional importance of 3'UTR sites and complements the findings of prior orthogonal assays (e.g., by eCLIP data, among others). The authors propose that these results will aid in genomic analysis but stop short of providing enough guidance (or tools) to aid in using these results to do that. The article is very well written and will be of interest to the human genomics field.

Major:

1. The authors state that 25% of genes in gnomAD have ≥ 1 3'UTR variant labeled as highly disruptive (lines 439-441) before highlighting two genes of interest. The manuscript would benefit from an aggregate genome-wide analysis of whether these $\sim 7,650$ genes contain genes that would not be expected to be tolerant of disruption? For example, are these depleted for essential genes, loss-of-function intolerant genes, haploinsufficient genes, and autosomal dominant disease associated genes, genes in which 3'UTR variants are reported to be pathogenic?

2. The authors specifically mention that the results of this study can inform variant interpretation as part of the Ellingford et al., framework (lines 570-574) intended for rare variants in Mendelian disease. While the study helpfully indicates functional regions of highest interest in the 3'UTR based on iMAPS, in the clinical setting, mapping the position of a patient's variant (which is often absent in gnomAD) to one of these high iMAPS regions would be challenging given the number and complexity of datasets that would need to be queried. Provision of the genome wide coordinates used to collate 3'UTR variants for each of the analyses (specifically the sites subsequently identified to be under moderate-strong negative selection) with the corresponding iMAPS scores, would add great value to the study.

3. Particularly missing (expanding on point 2) is how to apply these approaches to variants that are not in gnomAD. The iMAPS score does not apply there since variants are absent. But authors could still comment on how these scores can inform variants in these regions that are absent from gnomAD. In the Whiffin et al. manuscript that did a similar analysis of 5' UTR variation, the authors provided a VEP plug in (<https://imperialcardiogenetics.github.io/UTRannotator/>). Providing a tool like this would be optimal but otherwise providing additional supplemental files with regions of high interest for guiding analysis is needed.

4. In Table S2 the iMAPS scores are provided for three specific 3' UTR element types (ReP, miRNA, PAS) along with the corresponding variants present in gnomAD. Numerous abbreviations are not explained in the "classification" column, and it is currently not possible to link the analysis descriptions in the results to the table (e.g., "miRNA_vert_8mer_mult_prim", "ReP_both_min_0.05_dis").

Minor:

1. The authors nicely introduce the contribution of 3' UTR variants to GWAS hits in the context of

common disease (lines 57-61), a line should be added on the contribution of 3' UTR variants to rare Mendelian disease (i.e., according to the P/LP variants submitted to ClinVar) to also place the importance of their results in context for rare disease.

2. In lines 305-307, it would be useful to indicate the approximate number of genes, genome wide, that are regulated by Pumilio family binding sites.

3. In Figure 1c the use of overlapping colors in graph is not clear, consider different shapes.

4. In Figures 2-6 the tick marks denoting the iMAPS values of different variant classes (e.g., synonymous, missense) are not visible enough, consider a labeled dotted horizontal line across the graph or larger marks.

5. Consider adding additional description of the statistics in the figure legends.

6. Line 383 "on" duplicated

7. Figure 5B has the p-value written out and 5C has a star. Why the different style?

8. Line 595: not sure what "greater than 42 (4/3 of overall median)" means.

REVIEWER COMMENTS

Reviewer #1 (Remarks to the Author):

This manuscript describes a modification to the established MAPS (Mutability Adjusted Proportion Singleton) method, specifically designed to calibrate to the mutagenic propensities of 3' UTR, which may be different from those in coding regions. This is an important problem, as the functional significance of variants in 3' UTR are, without doubt, currently under-appreciated. The original MAPS corrects for the confounding effects of differential mutagenicity when assessing the evidence for negative selection. The authors introduce iMAPS, which performs the calibration based on intergenic regions broadly, though the method was applied here specifically to the interpretation of variants in 3' UTRs of genes. The authors appeared to also have stratified by varying amounts of sequence context, which seems appropriate. One question I have is whether the calibration could be improved by focusing specifically on pooled UTR rather than admitting all intergenic stretches, given that many gene loci are transcribed at substantially higher levels and may have different mutagenicity profiles than so-called intergenic "deserts".

We explored calibrating iMAPS using 3' UTR variants. To do so, we derived expected proportion singleton values for each possible base change + context using the same method we employed for iMAPS, but with 3' UTR variants rather than intergenic variants. Then, we compared the observed proportion singleton and expected proportion singleton values for all 4^5 (= 1,024) 5-mers in 3' UTRs. Under the premise that the vast majority of occurrences of the vast majority of all possible 5-mers are not under strong negative selection, better models of how non-selective forces influence proportion singleton values should explain more of the variance in the observed proportion singleton values.

When we fit linear models for our iMAPS approach and the 3' UTR comparison approach, we found that both approaches resulted in very similar correlations and residual standard errors, and similar residuals across 5-mers (Fig. R1). The performance of our intergenic iMAPS approach was notable given that the sets of variants that derived the expected and observed values for iMAPS are completely independent, while the 3' UTR approach utilized the same variants to derive both the expected proportion singleton and observed proportion singleton values. Collectively, these results suggest that there is not a widespread influence of transcription outside of what is already accounted for by our approach, as suggested by Fig. 1C. For comparison, see the newly added Supplementary Figure 1 comparing iMAPS and MAPS approaches.

Additionally, our iMAPS approach has some advantages over a potential 3' UTR calibration method: If a class of variants is frequently under strong negative selection, such signal can be easily detected using iMAPS, whereas calibration using the same variants one is interested in assessing (in this case 3' UTR) would mask such signal. Calibration using 3' UTR variants also sets the average negative selection across 3' UTRs to 0 or neutral. Since it is well established that 3' UTRs contain functional regulatory elements and are on average more conserved across species than intergenic regions, it is of more utility and more intuitive to use an approach such as iMAPS that can quantify the average negative selection in 3' UTRs relative to a more neutral region of the genome.

a

b

c

Figure R1. a) observed versus expected proportion singleton for all 5-mers in 3' UTRs using the iMAPS method. b) The same analysis as in "a" using 3' UTRs to obtain the expected proportion singleton values performs similarly to using independent intergenic regions as was done for "a". c) A scatter plot of the residuals for each 5-mer resulting from the regressions in "a" and "b" are roughly distributed along the diagonal, suggesting that these two approaches provide similarly accurate calibration and there is no major systemic improvement when calibrating using 3' UTRs.

The authors provide extensive analyses and make a strong case that their method uncovers clear signs of selection on known functional elements such as RNA-binding-protein binding sites, miRNA target sites, and polyadenylation signals, and observe different selection on constitutively versus alternatively included locations after cleavage and polyadenylation, consistent with recent results on the use of alternative polyadenylation sites in humans and their probable adaptive role. These nuanced analyses appear to follow quite well our current understanding of the biology of post-transcriptional gene regulation through mechanisms involving 3' UTR. Moreover, the authors demonstrate that their method is more specific than broad conservation signals quantified by PhastCons/PhyloP, which do not (though technically they could) parse their signals down to the level of individual nucleotide substitutions. They also make the important point that the evidence for recent adaptation in regulatory regions on the human lineage suggests alternative analytic approaches such as this one.

I do think this manuscript could benefit from a better description of the method for non-specialists (including computational non-specialists). In particular, I think it would be useful to more directly contrast differences in the mathematical formulation of the iMAPS and MAPS methods. My understanding is that the original MAPS was a regression model, whereas this appears to be a contingency table-based method.

We have added a “Quantifying negative selection” subsection of the Methods section that more clearly compares and contrasts our iMAPS approach to the earlier MAPS approach.

In summary, I think the problem addressed by the authors is potentially an extremely important one, given the importance of accurate variant interpretation in the context of disease studies and patient diagnosis, and the generally under-appreciated nature of gene regulation via 3' UTR. Extensive analyses by the authors convincingly argue that these types of modifications to existing approaches to variant interpretation in non-coding regions deserve attention.

Reviewer #2 (Remarks to the Author):

Authors introduced a new method to calculate the gene expression-related impact of non-coding variants in post-transcriptional level, mostly focused on negative selections of the variants by considering population-level germline variations. The method was named as iMAPS (intergenic mutability-adjusted proportion singleton), an improved method of the MAPS, with calibrating of observed singleton variant numbers with expected number of singleton variants using extended flanking k-mers. The major advances of this method compared to the previous method appear to be that 1) removal of singleton variants that associated with transcription to clearly capture the post-transcriptional signals and; 2) utilized orthogonal approaches to prioritize regulatory motifs in 3UTR ; 3) used intergenic region as control and 4) larger sequence context to calculate expected number of singleton variants (upto five in a side); 5) utilized the GnomAD database to estimate iMAPS of variants from 3'UTR contexts. With a naïve analysis, all variants in 3'UTR displayed lower iMAPS in eCLIP-RBP sites, miRNA sites, polyA signal, than synonymous mutations, with showing less negative selection pressure in 3'UTR than in coding regions. However, prioritizing the variants using orthogonal datasets (BIND-N-Seq.. and conservation), they found that some variants sets displayed greater negative selection than synonymous mutations, suggesting that the iMAPS method could unprecedentedly capture functional non-coding variants in 3'UTRs. Although the authors presented interesting and unprecedented reports, there are some descriptions and definitions that they have to clarify in interpreting the results as well as there are technical issues that they should be carefully addressed.

[Major comments]

1. Authors claimed that the iMAPS detected functional non-coding variants in 3'UTRs using orthogonal datasets and information, some of which displayed greater negative selection than synonymous mutations, however previous methods, such as MAPS and methods prioritizing variants on conservation scores, could still capture those same functional variants with lower signal though. I wonder how the scores are correlated among iMAPS and other previous methods.

2. Related to comment 1, I wonder if the iMAPS outperformed previous methods in terms of sensitivity and specificity. They need to carefully demonstrate the improvement of the suggested method using standard data to see whether they capture more true positives and less false positives.

3. I wonder the rationale of the use of intergenic region and larger sequence context for calculating expected number of singleton variants. Authors claimed the power of the use of them with the analysis of functional context of 3'UTR, but they failed to clearly show the improvement over conventional method when using authors' method. Rather they just prioritized functional variants using orthogonal datasets.

These three related comments are addressed below:

First, we would like to clarify that our iMAPS approach does not classify *individual* variants as functional / non-functional or pathogenic / non-pathogenic. We have used iMAPS to identify and validate *classes* of 3' UTR variants that are under strong negative selection, and have labeled variants belonging to these classes as “disruptive” in terms of their molecular impact. We think of this labelling as analogous to how non-synonymous CDS variants are considered disruptive for altering protein sequence, as articulated in the results section for Figure 6. Furthermore, while it would be highly desirable to have orthogonal gold-standard datasets for negative selection (or functionality in general) for non-coding variants in 3' UTRs, we are not aware of any suitable datasets that classify large numbers of such variants to make the proposed sensitivity and specificity analyses possible. We did make efforts to cross-validate our variant classification and stratification approaches using orthogonal datasets (e.g., eQTL, tamVar) where possible (e.g. Fig 2B, 2D, 3D).

We have compared the accuracy of our iMAPS approach more carefully to the earlier MAPS approach. To do so, we derived expected proportion singleton values obtained using either our iMAPS approach or the MAPS approach as described by Whiffin et al. (2020) for all $4^5 (= 1,024)$ 5-mers in 3' UTRs. We then compared these expected proportion singleton values to the actual proportion singleton values observed. Under the premise that the vast majority of occurrences of the vast majority of all possible 5-mers are not under strong negative selection in 3' UTRs, a better model of how non-selective forces influence proportion singleton values should explain more of the variance in the observed proportion singleton values. When we fit linear models for our iMAPS approach and the MAPS approach, iMAPS resulted in 1) higher correlations; 2) reduced residual standard error (over 1.8-fold lower); and 3) the majority of 5-mers with regression residuals closer to 0 relative to MAPS. These data suggest that iMAPS more accurately detected signals of negative selection in 3' UTRs by better accounting for non-selective forces that influence allele frequency spectra / proportion singleton values. These results have been added to the Supplementary Data as Supplementary Figure 1.

4. Moreover, they selected different number of flanking sequence context by each functional group of 3'UTR. However, it can make confound. For example, if we want to compare two sets where the used flanking sequence lengths are measured differently, it may produce different results than when comparing them separately. Further clarification on methodological justification is needed.

To clarify, the amount of sequence context used is independent of any functional group that a given 3' UTR variant may belong to, but is rather assigned based on flanking nucleotide context and other local genomic features. Please see the updated text in the Results section for Figure 1, and the newly added “Accounting for the impact of extended flanking sequence contexts” sub-section in the Methods section where we have added text to further clarify and justify our approach.

5. Synonymous mutations show their phenotypes through changes in mRNA stability, RNA structure, translation kinetics, and splicing. The negative selection for the synonymous mutations would seem to be varied (splicing could be greater than others), so if authors analyze synonymous mutations separately by the biological consequences, they could see different results.

Previous work assessing negative selection faced by non-coding regulatory variants in humans had focused on specific regions such as 5' UTRs (e.g. Whiffin et al., 2020). We focused on regulatory variation in 3' UTRs, assessing negative selection across multiple major modes of post-transcriptional gene regulation including miRNA binding, cleavage and polyadenylation, and other diverse RBP-RNA interactions. We suggest that future work can leverage our iMAPS method to detect signals of negative selection in other gene regions that each have their own distinct and complex gene-regulatory processes.

6. Throughout the study, they mostly tried to capture the computational association and enrichments to find functional non-coding variants with no experimental validation. If they deliver an experimental validation for core findings to show the functional non-coding variants are actual, it could strengthen their works further.

A massively parallel reporter assay can be used to assess the impact of individual short segments of 3' UTRs, including variant alleles, on steady-state transcript levels (that may be altered due to changes in transcript stability, etc.) in cells. We piloted such an experimental system, testing 28 ReP site-disrupting 3' UTR gnomAD variants. Some 39% (11/28) of the variants tested directly and reproducibly modulated steady-state transcript levels. These experimental results support our approaches to classify and interpret regulatory variation in 3' UTRs, and have been added to the Results section for Figure 2, and to the Supplementary Data as Supplementary Figure 5.

7. In some analysis, they focused on disrupting motifs but others focusing on the creation and strengthened motifs to prioritize functional variants without clear rationales. They need to clarify why they focused on one side.

Large-scale datasets such as eCLIP and polyA_DB are necessarily generated using a small number of cell lines / individuals. Since any individual genome is overall very similar to the reference genome, most signals detected are from reference alleles, and we don't have access to equivalent signals that might result from the many alternative alleles distributed across many individuals. Since reproducible eCLIP peaks and poly(A) sites from (primarily) reference alleles were readily available (and required to define high-confidence sites), we focused most of our analyses on disruption of these elements.

Conversely, short of generating extensive eCLIP / polyA_DB datasets across thousands of genetically diverse cell lines / individuals, there is no way to know what peaks or poly(A) sites might be created by a given alternative allele. While creation of functional elements by alternative alleles is of interest, we expected that such signals might be subtle and difficult to detect. We investigated alternative alleles that created putative binding sites for the very well understood Pumilio family of RBPs, reasoning that creation of a strong (low nM affinity) motif

for a potent RNA destabilizing factor might yield a signal, and found this to be the case in Figure 3. We have clarified our rationale for why we analyzed this case and why we mostly focused on motif disruption in the Results section related to Figure 3.

8. There are many RBPs in the analysis of figure 2C. Although the authors showed the negative effect of creation of PUM binding sites on the transcript levels in figure 3C, it would be very helpful to understand each protein function if the authors show the respective effect of RBP binding site deletion of each RBP on the transcript level, besides eQTL PIP changes.

Unfortunately there is not enough overlap between the ReP sites and available eQTL data to make such an analysis possible for individual RBPs.

9. ReP sites seem to be more relevant to capture the functional non-coding variants in 3'UTRs. I wonder if the ReP sites are more evolutionally conserved than eCLIP sites and non-ReP sites in eCLIP sites, and wonder if the disrupting ReP sites are less/more conserved than preserving sites. In addition, I wonder if the ReP sites are more negatively selected in core regions of 3'UTRs than in variable regions.

To determine if ReP sites are more evolutionarily conserved across species than eCLIP peaks, we selected a matched control eCLIP region for each ReP site. Relative to eCLIP peak regions, positions within ReP sites were up to 65% more likely to be conserved across species. This rate was higher for both higher affinity ReP sites, and for higher confidence conservation thresholds, consistent with our human selection findings. These results have been added to the Supplementary Data as Supplementary Figure. 3. Since conventional cross-species conservation scores (e.g. PhyloP) do not consider the impact of each possible base change at each position, we were unable to assess if ReP site-disrupting base changes are more or less conserved across species than ReP site-preserving base changes. It would be interesting to compare the rates of negative selection in core and variable regions of 3' UTRs. However, since ReP sites are partially defined by eCLIP peaks, which are enriched in more frequently expressed core 3' UTR regions, the number of ReP sites in variable 3' UTR regions is insufficient for a comparative analysis.

10. They found some association between negative selection and AU flanking of canonical miRNA 8-mer sites. I wonder if they also found the relationship within the sites along with flanking regions.

To test if there is an association between AU content of miRNA targets and the extent to which the target sites are under selection, we stratified variants into three groups: those in 8-mer targets with a low (0-3), medium (4) or high (5-8) number of A+U bases. Negative selection was slightly elevated in high AU targets, however it should be noted that there were very few variants for sites with more than 5 A+U bases, and none for sites with more than 6 A+U nt. Overall, no striking relationship was observed between A+U content of the target sites and negative selection (Fig. R2).

Figure R2: iMAPS for variants in 8-mer targets of microRNAs conserved across vertebrates, binned by target AU content: low (0-3), medium (4), or high (5-8).

[Minor]

1. Authors used different number of flanking sequences to calculate expected number of singleton variants according to variant group, but it is not indicated in paper. Authors should indicate these numbers.

The total number of 3' UTR variants calibrated using a given amount of nucleotide context is now included in the figure legend for Figure 1B.

2. Clarify the intended meaning of “calibration” clearly used in page 5 and throughout the manuscript.

The text has been updated to further improve clarity.

3. For general reader, they need to explain what is MAPS in the abstract briefly because they used iMAPS term first in the abstract.

Given the character limitations in the abstract, we feel that introducing iMAPS and MAPS together is reasonable here. Detailed comparisons between iMAPS and MAPS (including the new Supplementary Figure 1) are now available to readers in the main text.

Reviewer #3 (Remarks to the Author):

Summary of content

Findlay, Romo, and Burge develop a measure of non-coding region negative selection and report negative selection comparable to missense variants for groups of 3'UTR variants at specific sites (RBP binding sites, miRNA target sites, and polyadenylation signals). Overall, in a series of well done analyses calculating the negative selection metric “iMAPS”, the study confirms the functional importance of 3'UTR sites and complements the findings of prior orthogonal assays (e.g., by eCLIP data, among others). The authors propose that these results will aid in genomic analysis but stop short of providing enough guidance (or tools) to aid in using these results to do that. The article is very well written and will be of interest to the human genomics field.

Major:

1. The authors state that 25% of genes in gnomAD have ≥ 1 3'UTR variant labeled as highly disruptive (lines 439-441) before highlighting two genes of interest. The manuscript would benefit from an aggregate genome-wide analysis of whether these ~7,650 genes contain genes that would not be expected to be tolerant of disruption? For example, are these depleted for essential genes, loss-of-function intolerant genes, haploinsufficient genes, and autosomal dominant disease associated genes, genes in which 3'UTR variants are reported to be pathogenic?

As suggested, we analyzed genes with at least one 3' UTR gnomAD variant labeled as highly disruptive for any evidence of being depleted for essential genes, LoF-intolerant genes, etc. We found evidence of substantial depletion for genes with pathogenic 3' UTR variants in ClinVar. This finding supports our approaches to identify disruptive variants in 3' UTRs, suggesting that the highly disruptive variants identified are under selection in genes where 3' UTR function has been implicated in disease pathogenesis. Since data availability for 3' UTR variants is often dependent on exome capture primer design, we wanted to test if this result was simply due to an ascertainment bias in the ClinVar dataset. We found that our highly disruptive 3' UTR variants were not depleted for genes with benign 3' UTR variants in ClinVar, suggesting the

observed depletion was specific to evidence of pathogenicity. Notably, we did not find evidence for depletion of essential genes or any other related gene sets investigated. These gene sets are primarily identified by the phenotypic or evolutionary consequences of decreased or absent expression. It should be noted that many of the highly disruptive 3' UTR variants identified likely increase transcript levels by disrupting destabilizing elements (e.g. ReP site and miR target disruption), or by disruption of primary (often distal) poly(A) signals, resulting in proximal poly(A) site selection yielding shorter and often more stable 3' UTRs. While it may be true of some genes, it is not *necessarily* the expectation that *increasing* the expression of an essential, LoF-intolerant, etc. gene would be universally deleterious. We have added a description of the results of this analysis in the Results section for Figure 6 and as panel C in Figure 6.

2. The authors specifically mention that the results of this study can inform variant interpretation as part of the Ellingford et al., framework (lines 570-574) intended for rare variants in Mendelian disease. While the study helpfully indicates functional regions of highest interest in the 3'UTR based on iMAPS, in the clinical setting, mapping the position of a patient's variant (which is often absent in gnomAD) to one of these high iMAPS regions would be challenging given the number and complexity of datasets that would need to be queried. Provision of the genome wide coordinates used to collate 3'UTR variants for each of the analyses (specifically the sites subsequently identified to be under moderate-strong negative selection) with the corresponding iMAPS scores, would add great value to the study.

3. Particularly missing (expanding on point 2) is how to apply these approaches to variants that are not in gnomAD. The iMAPS score does not apply there since variants are absent. But authors could still comment on how these scores can inform variants in these regions that are absent from gnomAD. In the Whiffin et al. manuscript that did a similar analysis of 5' UTR variation, the authors provided a VEP plug in (<https://imperialcardiogenetics.github.io/UTRannotator/>). Providing a tool like this would be optimal but otherwise providing additional supplemental files with regions of high interest for guiding analysis is needed.

We agree that it would add great value to provide more guidance for interpretation of 3' UTR variants not present in gnomAD, particularly those potentially implicated in rare disease. To do so, we have now updated Supplementary Table 2 to include *all possible* SNVs for all of the disruptive classes of variants we identified with iMAPS ≥ 0.05 , providing an accessible and comprehensive resource for interpretation and prioritization of putative functional regulatory variants (including rare variants) in 3' UTRs.

4. In Table S2 the iMAPS scores are provided for three specific 3' UTR element types (ReP, miRNA, PAS) along with the corresponding variants present in gnomAD. Numerous abbreviations are not explained in the "classification" column, and it is currently not possible to link the analysis descriptions in the results to the table (e.g., "miRNA_vert_8mer_mult_prim", "ReP_both_min_0.05_dis").

We have added a "file description" sheet to Supplementary Table 2 that includes full descriptions of the variant categories and all columns in the table.

Minor:

1. The authors nicely introduce the contribution of 3' UTR variants to GWAS hits in the context of common disease (lines 57-61), a line should be added on the contribution of 3' UTR variants to rare Mendelian disease (i.e., according to the P/LP variants submitted to ClinVar) to also place the importance of their results in context for rare disease.

Great point. We have added some text to the introduction.

2. In lines 305-307, it would be useful to indicate the approximate number of genes, genome-wide, that are regulated by Pumilio family binding sites.

Using a relative affinity threshold of 0.25, we identified putative creation or strengthening of putative Pumilio binding sites by 3' UTR gnomAD variants across 2,658 protein coding genes. This number has been added to the Results section for Figure 3.

3. In Figure 1c the use of overlapping colors in graph is not clear, consider different shapes.

We have updated Fig. 1C to use different shapes instead of different colors.

4. In Figures 2-6 the tick marks denoting the iMAPS values of different variant classes (e.g., synonymous, missense) are not visible enough, consider a labeled dotted horizontal line across the graph or larger marks.

We preferred to limit the display to the axis as we consider these iMAPS values general benchmarks. We have substantially increased the size of the tick marks for better visibility.

5. Consider adding additional description of the statistics in the figure legends.

6. Line 383 “on” duplicated

Fixed

7. Figure 5B has the p-value written out and 5C has a star. Why the different style?

To reduce clutter, we opted to display P values for figure panels with a single statistical test, and use asterisks to denote $P < 0.05$ for panels with more than one statistical test. All P values are listed in Supplementary Table 1.

8. Line 595: not sure what “greater than 42 (4/3 of overall median)” means.

The text has been updated for clarity: “...regions with median sequencing coverage less than 25 (3/4 or 75% of the genome-wide median coverage of 32) or greater than 42 (4/3 or 133% of the genome-wide median coverage of 32)...”

REVIEWERS' COMMENTS

Reviewer #2 (Remarks to the Author):

All initially raised concerns and comments were now addressed, and no more further comments.

Reviewer #3 (Remarks to the Author):

The authors edits have addressed my concerns very well and this remains a manuscript that will of great interest and utility to the human genomics field in understanding functional 3'UTR variation.

I did want to note that the author's comment in the introduction of 1000 3'UTR variants in ClinVar – only 1 of which overlaps with their predicted sites based on my reading – may point to these variants being in the 3'UTR in the canonical transcript but maybe in a coding portion of the gene in another transcript. This might be worth checking before reporting. Alternatively, more caveats could be added – variants in the 3'UTR in the canonical transcript (may have additional annotations in other transcripts).

REVIEWERS' COMMENTS

Reviewer #2 (Remarks to the Author):

All initially raised concerns and comments were now addressed, and no more further comments.

Reviewer #3 (Remarks to the Author):

The authors edits have addressed my concerns very well and this remains a manuscript that will of great interest and utility to the human genomics field in understanding functional 3'UTR variation.

I did want to note that the author's comment in the introduction of 1000 3'UTR variants in ClinVar – only 1 of which overlaps with their predicted sites based on my reading – may point to these variants being in the 3'UTR in the canonical transcript but maybe in a coding portion of the gene in another transcript. This might be worth checking before reporting. Alternatively, more caveats could be added – variants in the 3'UTR in the canonical transcript (may have additional annotations in other transcripts).

We did exclude all CDS from this analysis. We suspect the number we obtained was higher than other estimates (e.g. 437 from <https://simple-clinvar.broadinstitute.org>) since we used polyA_DB to define 3' UTRs, many of which are substantially extended relative to widely used annotations. We have updated “close to 1,000” to “hundreds” in the introductory text.